# Towards Reliable Code-as-Policies: A Neuro-Symbolic Framework for Embodied Task Planning

**Sanghyun Ahn**[1], **Wonje Choi**[1], **Junyong Lee**[1], **Jinwoo Park**[2], **Honguk Woo**[1,2*]

[1]Department of Computer Science and Engineering, Sungkyunkwan University
[2]Department of Artificial Intelligence, Sungkyunkwan University
{shyuni5, wjchoi1995, ljy7488, pjw971022, hwoo}@skku.edu

## Abstract

Recent advances in large language models (LLMs) have enabled the automatic generation of executable code for task planning and control in embodied agents such as robots, demonstrating the potential of LLM-based embodied intelligence. However, these LLM-based code-as-policies approaches often suffer from limited environmental grounding, particularly in dynamic or partially observable settings, leading to suboptimal task success rates due to incorrect or incomplete code generation. In this work, we propose a neuro-symbolic embodied task planning framework that incorporates explicit symbolic verification and interactive validation processes during code generation. In the validation phase, the framework generates exploratory code that actively interacts with the environment to acquire missing observations while preserving task-relevant states. This integrated process enhances the grounding of generated code, resulting in improved task reliability and success rates in complex environments. We evaluate our framework on RLBench and in real-world settings across dynamic, partially observable scenarios. Experimental results demonstrate that our framework improves task success rates by 46.2% over Code as Policies baselines and attains over 86.8% executability of task-relevant actions, thereby enhancing the reliability of task planning in dynamic environments.

## 1 Introduction

Recent advances in embodied control have leveraged large language models (LLMs) to enable flexible, instruction following, effectively bridging natural language understanding with executable actions in physical environments. For instance, SayCan [1] combines LLM-based task interpretation with a reinforcement learning (RL) affordance model to construct a hybrid policy that grounds high-level language instructions, such as "bring me the sponge." into sequences of low-level, predefined robotic skills. Building on this foundation, subsequent approaches have explored more expressive and compositional modes of action specification through code generation, introducing the paradigm of code-as-policies [2, 3, 4], where LLMs directly generate executable code to control embodied agents. This shift enables task planning that is more modular, interpretable, and adaptable to diverse environments, highlighting the potential of LLMs as general-purpose planners for robotic control.

While LLM-based code-as-policies approaches have demonstrated promising capabilities in fully observable and well-structured settings, their reliability deteriorates in dynamic or partially observable environments, where perceptual input is often sparse, delayed, or ambiguous. These limitations lead to incorrect or incomplete code generation, ultimately resulting in suboptimal task performance. For example, attempting to grasp a fragile object without access to accurate depth or height estimation may lead to dropping or damaging the object, preventing task completion. These challenges underscore

---

*Corresponding author

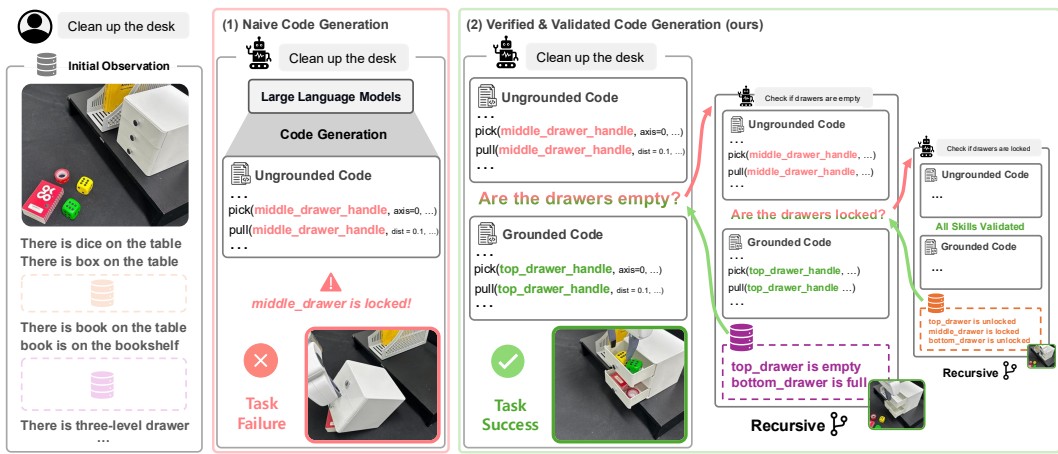

Figure 1: Concept of our NESYRO framework illustrated with an example of a room-cleaning task where drawer states are initially unknown. While (1) naive code generation fails without detecting missing observations, (2) NESYRO recursively probes the environment to recover drawer states, enabling the generation of grounded code that successfully completes the task.

the critical need for embodied agents to explicitly reason about uncertainty through exploratory yet safe interactions, and to verify the correctness of generated code prior to execution.

To address these challenges, we propose NESYRO, a neuro-symbolic robot task planning framework that incorporates explicit symbolic verification and interactive validation processes during code generation. Drawing inspiration from the long-standing software engineering principle of verification and validation (V&V) [5, 6, 7], our framework distinguishes between two key processes: verification ensures that the generated code is logically consistent and satisfies symbolic preconditions, while validation assesses whether the code is suitable for the current environment and task objectives. Specifically, symbolic verification statically checks code correctness using domain-specific symbolic tools, whereas interactive validation enables the agent to actively explore its environment to resolve ambiguities and acquire missing observations before task-specific execution.

Our NESYRO framework operates through a recursive composition of two phases: (i) Neuro-symbolic Code Verification, and (ii) Neuro-symbolic Code Validation. Following symbolic verification for code correctness, the interactive validation phase grounds each skill by identifying preconditions and invoking exploratory actions that establish those preconditions as effects, thereby transforming the environment state to enable the intended skill. This process resembles a form of backtracking search, where the agent navigates the environment to construct a valid execution path, progressively verifying and validating the code based on current observations and feedback from symbolic tools.

Figure 1 shows the concept of our framework through an object relocation task, comparing a naive code generation method and our approach. The naive method attempts to move an object without adequately accounting for uncertain factors, which results in the execution of actions that prevent task success. In contrast, our framework identifies the need for exploratory actions, ensures they are conducted safely, and successfully completes the task without causing damage.

We evaluate NESYRO on four task categories, including object relocation, object interaction, auxiliary manipulation, and long-horizon tasks, using both the RLBench [8] simulation and real-world settings. Experimental results demonstrate that NESYRO improves task success rate by 46.2% over the state-of-the-art baseline, Code as Policies [2], while achieving over 86.8% executability of task-relevant actions in real-world settings. These underscore the enhanced reliability of our framework for robust task planning in dynamic, partially observable environments

The contributions of this work are summarized as follows:

- We present the NESYRO framework to enable the automatic generation of executable code for task planning in dynamic, partially observable environments.
- We propose a novel recursive mechanism that combines symbolic verification and interactive validation to actively infer and satisfy task-relevant preconditions through exploratory code execution.

- Extensive evaluations on RLBench and real-world tasks demonstrate that NESYRO significantly improves both task success rates and the executability of task-relevant skills.

## 2 Related work

**LLM-based embodied control.** In the field of embodied control, there is an emerging trend of utilizing LLMs for reasoning and planning tasks [1, 9, 10, 11, 12, 13, 14, 15, 16, 3]. Building on the high-level reasoning capabilities of LLMs, recent approaches have explored generating executable code as a direct control policy, a paradigm often referred to as code-as-policies [2, 4, 17, 18, 19]. Rather than mapping instructions to predefined skills or discrete action primitives, these methods prompt LLMs to generate Python-like scripts that can be directly executed by embodied agents such as robots. This demonstrates that LLMs are capable of synthesizing low-level control logic, enabling greater flexibility and generalization across a diverse range of tasks. Yet, in dynamic or partially observable settings, the generated code often lacks proper grounding, resulting in incomplete or non-executable outputs. To mitigate this, NESYRO enhances the environmental grounding and reliability of generated code by integrating explicit feedback into the code generation process.

**Code verification and validation.** Verification and validation are foundational techniques in software engineering for ensuring the correctness and robustness of programs. Verification typically involves static analysis methods such as formal verification, theorem proving, and model checking [20, 21, 22, 23, 24], aiming to prove that a program satisfies its specification before execution. Validation assesses runtime behavior through unit testing, integration testing, system-level evaluation, and runtime verification [7, 25, 6, 26, 27], ensuring that the code performs as intended under real-world conditions. Recent works have explored combining these principles with LLMs to improve code reliability via various forms of static analysis and runtime feedback [28, 29, 5]. Still, existing approaches are limited to static or simulated settings and lack grounding in real-world environments, which is an essential requirement for embodied agents. Our framework addresses this by enabling agents to identify missing task-relevant observations in dynamic and partially observable environments.

**Neuro-symbolic system.** Recent neuro-symbolic systems combine the generalization capabilities of LLMs with the robustness and interpretability of symbolic reasoning tools. This hybrid approach has been actively investigated in areas such as symbolic problem solving, planning, and program synthesis [30, 31, 32, 33, 34, 35, 36]. Neuro-symbolic approaches in embodied agents commonly employ LLMs for perception and natural language instruction understanding, while utilizing symbolic tools to perform high-level task planning [37, 38, 39, 40, 41]. However, existing neuro-symbolic agents rely on fixed modular structures or pre-defined procedures, limiting their adaptability to missing observations and environmental uncertainty. NESYRO integrates symbolic reasoning with interactive validation and exploratory interactions, enabling reliable task planning in dynamic environments.

## 3 NESYRO Framework

### 3.1 Problem Formulation

We tackle the automatic generation of executable code for task planning and control in embodied agents operating in dynamic, partially observable environments. The environment is modeled as a Partially Observable Markov Decision Process (POMDP) $\mathcal{M} = (\mathcal{S}, \mathcal{A}, \mathcal{G}, T, R, \Omega, \mathcal{O})$ [42, 43, 14], where $s \in \mathcal{S}$ is a state, $a \in \mathcal{A}$ an action, and $g \in \mathcal{G}$ is a high-level goal (e.g., "pick up the red mug"). $T : \mathcal{S} \times \mathcal{A} \to \mathcal{S}$ is the transition function describing dynamics. The reward function $R : \mathcal{S} \times \mathcal{A} \times \mathcal{G} \to \{0, 1\}$ returns a binary success signal, which is common in robotics where only task completion is observable. Due to partial observability, observations $o \in \Omega$ are received via $\mathcal{O} : \mathcal{S} \times \mathcal{A} \to \Omega$, where observations are represented in symbolic form, composed of structured predicate-based expressions (e.g., `is_locked(drawer)`, `on(object, surface)`). Under the code-as-policies paradigm, the LLM generates policy code $\pi$ from observation history and goal as input and internally encodes the actions necessary to complete the task. When executed via tool exe, it yields policy $\text{exe}(\pi)$. The set of policy codes is $\Pi = \{\pi \mid \text{exe}(\pi) : \Omega^* \to \mathcal{A}\}$, where $\Omega^*$ is the set of all finite observation histories $o_{\leq t} = (o_0, \ldots, o_t)$ with each $o_i \in \Omega$. Our goal is to find the policy code $\pi$ that maximizes the expected return:

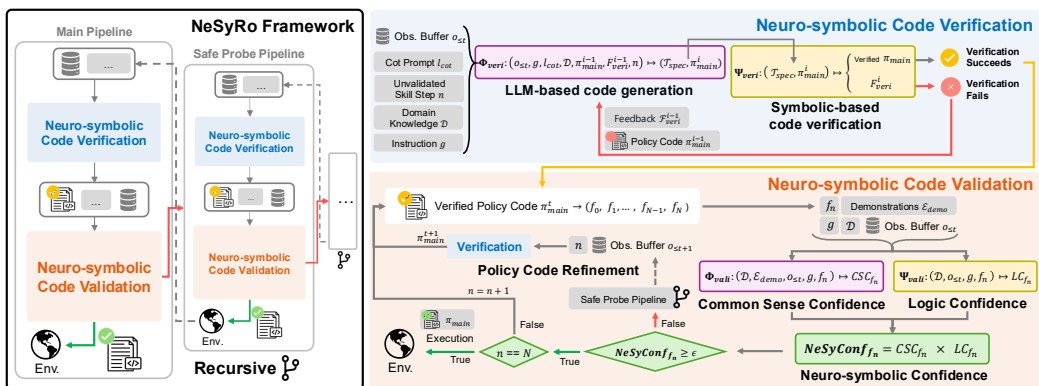

Figure 2: The NESYRO framework with *Neuro-Symbolic Code Verification* and *Neuro-Symbolic Code Validation* phases. It recursively verifies and validates the policy code, while incrementally acquiring observations.

$$\pi^* = \operatorname*{argmax}_{\pi \in \Pi} \; \mathbb{E}_{g \sim \mathcal{G}, \tau \sim \mathcal{P}(\mathrm{exe}(\pi), g)} \left[ \sum_{t=0}^{\infty} R(s_t, \mathrm{exe}(\pi)(o_{\leq t}), g) \right]. \tag{1}$$

Here, $\tau = (s_0, o_0, a_0, s_1, o_1, a_1, \dots)$ denotes the trajectory generated by executing $\mathrm{exe}(\pi)$ in the environment, and $\mathcal{P}(\mathrm{exe}(\pi), g)$ is the resulting trajectory distribution induced by $\mathrm{exe}(\pi)$ under $g$, $T$, and $\mathcal{O}$. In our implementation, each action $a_t$ in $\tau$ corresponds to a skill function composed of multiple low-level control APIs encoded within $\pi$. Since $\mathcal{M}$ is partially observable, $\pi^*$ must balance exploration (to reduce uncertainty) and exploitation (to achieve goals), ensuring reliable task planning in dynamic environments.

## 3.2 Overall Framework

To achieve the objective described in Eq. (1), we introduce NESYRO, designed to achieve the generation of executable and grounded code through dynamic reconfiguration of reasoning components. As illustrated in Figure 2, NESYRO operates in two key phases: Phase i), *Neuro-Symbolic Code Verification*, which ensures the logical correctness of the policy code with respect to the generated task specification; and Phase ii), *Neuro-Symbolic Code Validation*, which ensures environmental feasibility by evaluating and refining skills based on their grounding.

In the verification phase i), given a language instruction $g$ and current observation $o_{\leq t}$, the LLM generates a task specification $\mathcal{T}_{\mathrm{spec}}$ along with the initial policy code $\pi_{\mathrm{main}}$. The symbolic tool then verifies whether $\pi_{\mathrm{main}}$ satisfies $\mathcal{T}_{\mathrm{spec}}$. If verification fails, the symbolic tool provides feedback to the LLM, which iteratively refines $\pi_{\mathrm{main}}$ until a verified version is obtained. In the validation phase ii), the sequence of skills defined in $\pi_{\mathrm{main}}$ is validated sequentially using a neuro-symbolic confidence score, NeSyConf, which integrates symbolic feasibility and commonsense plausibility. If a skill's confidence score falls below a threshold $\epsilon$, NESYRO synthesizes a safe probe policy code $\pi_{\mathrm{probe}}$ to recover missing observations. $\pi_{\mathrm{probe}}$ is recursively processed through the composition of the verification and validation phases until all skills are grounded.

This recursive structure induces a policy tree rooted at $\pi_{\mathrm{main}}$, where each $\pi_{\mathrm{probe}}$ serves as a subroutine that enables successful validation of its parent. The recursive process continues until all required observations have been acquired and every skill in $\pi_{\mathrm{main}}$ is validated. The final output is a grounded version of $\pi_{\mathrm{main}}$, aligned with both $\mathcal{T}_{\mathrm{spec}}$ and the current environment.

## 3.3 Neuro-symbolic Code Verification

**LLM-based code generation.** Given a language instruction $g$ and observation $o_{\leq t}$, a verification LLM, denoted as $\Phi_{\mathrm{veri}}$, is prompted to reason in a chain-of-thought (CoT) manner [44, 45], synthesizing key objectives and constraints into a task specification $\mathcal{T}_{\mathrm{spec}}$. It then uses this specification to generate the policy code $\pi_{\mathrm{main}}^i$, for instance in Python, defining a sequence of skills along with their

parameters and required libraries.

$$\Phi_{\text{veri}} : (o_{\leq t}, g, l_{\text{cot}}, \mathcal{D}, \pi_{\text{main}}^{i-1}, \mathcal{F}_{\text{veri}}^{i-1}, n) \mapsto (\mathcal{T}_{\text{spec}}, \pi_{\text{main}}^{i}) \tag{2}$$

Here, $o_{\leq t}$ is the current observation, initially from $o_0$ and incrementally updated via probe. $l_{\text{cot}}$ is the CoT prompt guiding the $\Phi_{\text{veri}}$ to generate the specification as an intermediate step. $\mathcal{D}$ denotes domain knowledge, consisting of available skills represented as parameterized function calls, each defined by its applicability conditions and resulting effects, as well as object types and attributes that map these skills to the environment. $\mathcal{F}_{\text{veri}}^{i-1}$ is verification feedback from the previous iteration, used by the LLM to generate the revised $\pi_{\text{main}}^{i}$. Importantly, $\pi_{\text{main}}^{i-1}$ and $\mathcal{F}_{\text{veri}}^{i-1}$ are provided only when the previous verification attempt has failed. The index $n$ indicates the skill function call order in $\pi_{\text{main}}$ from which the code refinement begins, while calls prior to $n$ remain unchanged. When $n=0$, it corresponds to the initial code generation. The resulting $\mathcal{T}_{\text{spec}}$ captures the high-level intent, constraints, and relevant subgoals derived from the $g$ and $o_{\leq t}$. $\pi_{\text{main}}^{i}$ is then passed to the symbolic verification tool.

**Symbolic-based code verification.** Next, a symbolic verification tool $\Psi_{\text{veri}}$ (i.e., SMT solver) checks whether $\pi_{\text{main}}^{i}$ satisfies $\mathcal{T}_{\text{spec}}$, identifying any violations of constraints defined in the specification.

$$\Psi_{\text{veri}} : (\mathcal{T}_{\text{spec}}, \pi_{\text{main}}^{i}) \mapsto \begin{cases} \text{verified } \pi_{\text{main}}, & \text{if verification succeeds} \\ \mathcal{F}_{\text{veri}}^{i}, & \text{if verification fails} \end{cases} \tag{3}$$

If verification fails, $\Psi_{\text{veri}}$ provides detailed $\mathcal{F}_{\text{veri}}^{i}$ that identifies the specific parts of $\pi_{\text{main}}^{i}$ violating $\mathcal{T}_{\text{spec}}$, such as incorrect parameter bindings or structural mismatches. This feedback is then passed to the next $\Phi_{\text{veri}}$ iteration to generate a revised $\pi_{\text{main}}^{i}$. Once $\pi_{\text{main}}^{i}$ passes verification, resulting in a verified version of $\pi_{\text{main}}$, we proceed to the *Neuro-Symbolic Code Validation* phase.

## 3.4 Neuro-symbolic Code Validation

The verified policy code $\pi_{\text{main}}$ is parsed into a sequence of skill function calls, $\pi_{\text{main}} = (f_0, f_1, \ldots, f_N)$, where $N$ denotes the maximum skill step. Unlike the *Neuro-symbolic code verification* phase, which reasons over the entire $\pi_{\text{main}}$ holistically, the validation process evaluates and refines each skill sequentially to assess its feasibility in the current environment. The index $n$, as in Eq. (2), denotes the skill step under validation and represents the first unvalidated step in $\pi_{\text{main}}$.

**Neuro-symbolic confidence score.** To assess skill feasibility, we introduce Neuro-Symbolic Confidence score (NeSyConf), which combines Common Sense Confidence (CSC) from a validation LLM denoted $\Phi_{\text{vali}}$ and Logic Confidence (LC) from a symbolic validation tool $\Psi_{\text{vali}}$ in parallel.

$$\Phi_{\text{vali}} : (\mathcal{D}, \mathcal{E}_{\text{demo}}, o_{\leq t}, g, f_n) \mapsto \text{CSC}_{f_n} \tag{4}$$

The $\text{CSC}_{f_n}$ estimates the likelihood that a given skill $f_n \in \pi_{\text{main}}$ will succeed under the current observation $o_{\leq t}$ and instruction $g$, based on both domain knowledge $\mathcal{D}$ and retrieved demonstrations $\mathcal{E}_{\text{demo}}$. To compute this, we insert the code snippet corresponding to $f_n$ into the LLM prompt along with $\mathcal{D}$, $o_{\leq t}$, $g$, and $\mathcal{E}_{\text{demo}}$. $\Phi_{\text{vali}}$ then assigns token-wise probabilities to the $f_n$, and we compute a perplexity-based score to estimate the skill's plausibility. The cumulative log probabilities are normalized to produce a consistent confidence $\text{CSC}_{f_n}$. To reduce hallucinations and improve estimation accuracy, we retrieve skill-level demonstrations $\mathcal{E}_{\text{demo}}$ whose contexts closely resemble the current situation and include them in the prompt as guidance.

$$\Psi_{\text{vali}} : (\mathcal{D}, o_{\leq t}, g, f_n) \mapsto \text{LC}_{f_n} \tag{5}$$

The logic-based confidence $\text{LC}_{f_n}$ is computed by the symbolic validation tool (i.e., PDDL planner), which assesses whether the $f_n$ is symbolically feasible under the $o_{\leq t}$, $g$, and $\mathcal{D}$. We use a $\Psi_{\text{vali}}$ to check whether $f_n$'s preconditions hold in $o_{\leq t}$. If the planner successfully generates a plan including $f_n$, we set $\text{LC}_{f_n} = 1$; otherwise, we set $\text{LC}_{f_n} = 0$, indicating symbolic infeasibility under the $o_{\leq t}$.

$$\text{NeSyConf}_{f_n} = \text{CSC}_{f_n} \times \text{LC}_{f_n} \text{ with: } \begin{cases} \text{proceed to NeSyConf}_{f_{n+1}}, & \text{if NeSyConf}_{f_n} \geq \epsilon \\ \text{generate } \pi_{\text{probe}} \text{ using } \mathcal{F}_{\text{csc}}, \mathcal{F}_{\text{lc}} & \text{otherwise} \end{cases} \tag{6}$$

The $\text{NeSyConf}_{f_n}$ represents the final confidence score for $f_n$, computed by multiplying $\text{CSC}_{f_n}$ and $\text{LC}_{f_n}$. This score estimates whether $f_n$ is correctly grounded in the environment and likely to succeed upon execution. If $\text{NeSyConf}_{f_n} < \epsilon$, our framework initiates a safe probe policy code $\pi_{\text{probe}}$ using feedback from each component, namely $\mathcal{F}_{\text{csc}}$ and $\mathcal{F}_{\text{lc}}$.

**Safe probe.** If a $f_n$ receives a low confidence score, our framework responds by generating a safe probe policy code $\pi_{\text{probe}}$ to recover missing observations. Constructed using $\mathcal{F}_{\text{csc}}$ from CSC and $\mathcal{F}_{\text{lc}}$ from LC, $\pi_{\text{probe}}$ undergoes the same verification and validation process as $\pi_{\text{main}}$. Because $\pi_{\text{probe}}$ is validated before execution, the framework ensures that only safe and grounded code is deployed. This recursive structure generates a policy tree rooted at $\pi_{\text{main}}$, where each $\pi_{\text{probe}}$ functions as a subroutine that enables successful validation of its parent skill. Once $\pi_{\text{probe}}$ is executed, it collects new observations and updates the current observation to $o_{\leq t+1}$. This updated observation is then used in the subsequent *Policy code refinement* process to update $\pi_{\text{main}}$.

**Policy code refinement.** Following safe probe, the $o_{\leq t+1}$ is used to refine $\pi_{\text{main}}$ at the skill level. Specifically, instead of regenerating the entire policy, our framework targets the current $f_n$ and prompts the LLM to regenerate only its code segment using the $o_{\leq t+1}$. This regeneration is conducted through our *Neuro-symbolic Code Verification* process. The updated code for $f_n$ is then evaluated using Eq. (6), where its confidence score $\text{NeSyConf}_{f_n}$ is reassessed. This process of refinement and safe probe is repeated until $\text{NeSyConf}_{f_n} \geq \epsilon$. Once all skills $f_n \in \pi_{\text{main}}$ have been successfully validated, the grounded $\pi_{\text{main}}$ is executed.

Further implementation details, and algorithmic pseudocode are provided in Appendix A.

# 4 Experiment

## 4.1 Experiment Setting

**Environments.** We conducted experiments in both RLBench [8] and real-world settings using a 7-DoF Franka Emika Research 3 robotic arm, enabling reproducible evaluations via randomized initial states and instructions to analyze safe probe strategies in dynamic, partially observable scenarios. In contrast, real-world experiments evaluated robustness and generalizability under real-world noise and variability. In dynamic, partially observable scenarios, we defined four observability levels based on initial observation availability: *High Incompleteness* condition removes more than half of the essential observations, constraining the task-solving process. *Low Incompleteness* condition retains most observations, though the observation remains partially incomplete. *Stochastic Incompleteness* condition provides a randomly selected subset of observations, with the incompleteness level varying across episodes. Finally, *Complete* condition offers full relevant observations, rendering probe unnecessary. We denote these four levels as *High*, *Low*, *Stochastic*, and *Complete*, respectively, each evaluated over ten randomized trials with varied initial conditions and instructions.

Table 1: Task types and their corresponding probe types

| Task Type | Probe Type |
|---|---|
| *Object Relocation*
(e.g., moving tomatoes on a plate) | Robot Pose Adjust
(e.g., verifying which item is a tomato) |
| *Object Interaction*
(e.g., opening a drawer) | Object State Check
(e.g., checking whether a drawer is locked) |
| *Auxiliary Manipulation*
(e.g., opening a drawer in a dark room) | Object State Change
(e.g., turning on the light to locate the drawer) |
| *Long-Horizon*
(e.g., placing a tomato inside a drawer) | Uses two or more of the above probe types depending
on task structure and uncertainty |

**Task and probe types.** In dynamic, partially observable environments, tasks often require acquiring missing observations before execution. To support structured analysis of the probe, we define task types based on manipulation goals and missing observation roles. Specifically, we distinguish whether the uncertainty concerns object identity, object state, or auxiliary conditions. These distinctions define three task types, each reflecting a distinct observation-seeking pattern. In addition, *long-horizon* tasks involve multi-step goals. Correspondingly, we define three probe types to characterize observation acquisition: (1) Robot Pose Adjust, adjusting viewpoint to resolve ambiguity; (2) Object State Check, identifying hidden object states relevant to the task; and (3) Object State Change, performing auxiliary skills to enable observation of otherwise inaccessible states. These categories are functionally defined

Table 2: Task performance under varying levels of observability incompleteness in RLBench

| Methods | High | | Low | | Stochastic | | Complete | |
|---|---|---|---|---|---|---|---|---|
| | SR | GC | SR | GC | SR | GC | SR | GC |
| **Task Type: Object Relocation** | | | | | | | | |
| CaP | $25.0_{\pm7.1}$ | $41.5_{\pm8.8}$ | $30.0_{\pm0.0}$ | $43.8_{\pm1.8}$ | $10.0_{\pm0.0}$ | $36.3_{\pm1.8}$ | $90.0_{\pm0.0}$ | $92.5_{\pm3.5}$ |
| CaP w/ Lemur | $25.0_{\pm7.1}$ | $43.8_{\pm5.3}$ | $30.0_{\pm0.0}$ | $43.8_{\pm1.8}$ | $10.0_{\pm0.0}$ | $36.3_{\pm1.8}$ | $90.0_{\pm0.0}$ | $96.3_{\pm1.8}$ |
| CaP w/ CodeSift | $55.0_{\pm7.1}$ | $72.5_{\pm3.5}$ | $50.0_{\pm0.0}$ | $57.5_{\pm3.5}$ | $40.0_{\pm0.0}$ | $52.5_{\pm3.5}$ | $95.0_{\pm7.1}$ | $95.0_{\pm7.1}$ |
| LLM-Planner | $30.0_{\pm0.0}$ | $35.0_{\pm7.1}$ | $50.0_{\pm0.0}$ | $58.8_{\pm5.3}$ | $30.0_{\pm0.0}$ | $43.8_{\pm5.3}$ | $80.0_{\pm0.0}$ | $88.8_{\pm5.3}$ |
| AutoGen | $30.0_{\pm0.0}$ | $35.0_{\pm7.1}$ | $55.0_{\pm7.1}$ | $60.0_{\pm7.1}$ | $40.0_{\pm14.1}$ | $47.5_{\pm10.6}$ | $85.0_{\pm7.1}$ | $87.5_{\pm10.6}$ |
| NESYRO | $70.0_{\pm14.1}$ | $72.5_{\pm10.6}$ | $75.0_{\pm7.1}$ | $87.5_{\pm3.5}$ | $65.0_{\pm7.1}$ | $75.0_{\pm0.0}$ | $95.0_{\pm7.1}$ | $97.5_{\pm3.5}$ |
| **Task Type: Object Interaction** | | | | | | | | |
| CaP | $20.0_{\pm14.1}$ | $35.0_{\pm7.1}$ | $25.0_{\pm7.1}$ | $40.0_{\pm3.5}$ | $35.0_{\pm7.1}$ | $51.3_{\pm5.3}$ | $75.0_{\pm7.1}$ | $77.5_{\pm7.1}$ |
| CaP w/ Lemur | $35.0_{\pm7.1}$ | $47.5_{\pm7.1}$ | $35.0_{\pm7.1}$ | $47.5_{\pm3.5}$ | $30.0_{\pm14.1}$ | $46.3_{\pm12.1}$ | $85.0_{\pm7.1}$ | $86.3_{\pm8.8}$ |
| CaP w/ CodeSift | $40.0_{\pm0.0}$ | $65.0_{\pm7.1}$ | $50.0_{\pm14.1}$ | $55.0_{\pm7.1}$ | $40.0_{\pm0.0}$ | $60.0_{\pm14.1}$ | $90.0_{\pm14.1}$ | $90.0_{\pm14.1}$ |
| LLM-Planner | $5.0_{\pm7.1}$ | $15.0_{\pm0.0}$ | $40.0_{\pm14.1}$ | $53.8_{\pm5.3}$ | $35.0_{\pm7.1}$ | $42.5_{\pm14.1}$ | $55.0_{\pm7.1}$ | $63.8_{\pm8.8}$ |
| AutoGen | $40.0_{\pm0.0}$ | $48.8_{\pm1.8}$ | $50.0_{\pm0.0}$ | $58.8_{\pm5.3}$ | $50.0_{\pm0.0}$ | $57.5_{\pm0.0}$ | $75.0_{\pm7.1}$ | $76.3_{\pm8.8}$ |
| NESYRO | $70.0_{\pm0.0}$ | $76.3_{\pm1.8}$ | $80.0_{\pm0.0}$ | $83.8_{\pm1.8}$ | $70.0_{\pm14.1}$ | $73.8_{\pm8.8}$ | $90.0_{\pm0.0}$ | $92.5_{\pm0.0}$ |
| **Task Type: Auxiliary Manipulation** | | | | | | | | |
| CaP | $25.0_{\pm7.1}$ | $25.0_{\pm7.1}$ | $50.0_{\pm0.0}$ | $51.3_{\pm1.8}$ | $40.0_{\pm0.0}$ | $45.8_{\pm8.3}$ | $85.0_{\pm7.1}$ | $90.0_{\pm4.7}$ |
| CaP w/ Lemur | $30.0_{\pm14.1}$ | $30.0_{\pm14.1}$ | $50.0_{\pm14.1}$ | $58.3_{\pm7.1}$ | $30.0_{\pm14.1}$ | $34.2_{\pm15.3}$ | $85.0_{\pm7.1}$ | $90.8_{\pm3.5}$ |
| CaP w/ CodeSift | $5.0_{\pm7.1}$ | $5.0_{\pm7.1}$ | $55.0_{\pm7.1}$ | $57.5_{\pm3.5}$ | $35.0_{\pm7.1}$ | $35.0_{\pm7.1}$ | $90.0_{\pm0.0}$ | $93.3_{\pm0.0}$ |
| LLM-Planner | $15.0_{\pm7.1}$ | $15.0_{\pm7.1}$ | $30.0_{\pm0.0}$ | $37.5_{\pm3.5}$ | $10.0_{\pm14.1}$ | $15.0_{\pm7.1}$ | $75.0_{\pm0.0}$ | $80.0_{\pm0.0}$ |
| AutoGen | $15.0_{\pm7.1}$ | $15.0_{\pm7.1}$ | $35.0_{\pm7.1}$ | $40.0_{\pm0.0}$ | $20.0_{\pm0.0}$ | $22.5_{\pm3.5}$ | $80.0_{\pm0.0}$ | $80.0_{\pm0.0}$ |
| NESYRO | $60.0_{\pm0.0}$ | $80.8_{\pm1.2}$ | $70.0_{\pm14.1}$ | $74.2_{\pm13.0}$ | $70.0_{\pm14.1}$ | $85.8_{\pm5.9}$ | $95.0_{\pm7.1}$ | $96.7_{\pm4.7}$ |
| **Task Type: Long-Horizon** | | | | | | | | |
| CaP | $0.0_{\pm0.0}$ | $0.0_{\pm0.0}$ | $20.0_{\pm0.0}$ | $40.4_{\pm6.6}$ | $0.0_{\pm0.0}$ | $0.7_{\pm1.0}$ | $40.0_{\pm14.1}$ | $53.8_{\pm3.4}$ |
| CaP w/ Lemur | $0.0_{\pm0.0}$ | $0.0_{\pm0.0}$ | $30.0_{\pm0.0}$ | $47.1_{\pm0.0}$ | $0.0_{\pm0.0}$ | $1.6_{\pm0.2}$ | $55.0_{\pm7.1}$ | $67.1_{\pm5.1}$ |
| CaP w/ CodeSift | $0.0_{\pm0.0}$ | $0.0_{\pm0.0}$ | $30.0_{\pm14.1}$ | $45.8_{\pm7.9}$ | $5.0_{\pm7.1}$ | $5.0_{\pm7.1}$ | $65.0_{\pm7.1}$ | $71.4_{\pm6.7}$ |
| LLM-Planner | $0.0_{\pm0.0}$ | $0.0_{\pm0.0}$ | $10.0_{\pm0.0}$ | $11.4_{\pm0.0}$ | $5.0_{\pm0.0}$ | $12.9_{\pm8.1}$ | $35.0_{\pm7.1}$ | $44.4_{\pm9.9}$ |
| AutoGen | $0.0_{\pm0.0}$ | $5.5_{\pm3.0}$ | $30.0_{\pm14.1}$ | $39.2_{\pm10.3}$ | $20.0_{\pm0.0}$ | $28.5_{\pm7.2}$ | $50.0_{\pm0.0}$ | $55.1_{\pm0.8}$ |
| NESYRO | $45.0_{\pm7.1}$ | $65.2_{\pm6.7}$ | $45.0_{\pm7.1}$ | $58.1_{\pm6.1}$ | $35.0_{\pm7.1}$ | $41.9_{\pm8.1}$ | $65.0_{\pm7.1}$ | $73.7_{\pm8.3}$ |

and implemented for distinct behavioral purposes, thereby enabling generalization to varied settings. We describe the task types and their associated probe types in Table 1.

**Evaluation metrics.** To assess the objectives in Section 3.1, we adopt metrics from prior work [46, 11, 47, 48]. Success Rate (SR) assigns 100% for full task completion and 0% otherwise. Goal Condition (GC) measures the percentage of sub-goals achieved. In real-world experiments, we report Irreversible Actions (IA), counting irreversible actions during task execution.

**Baselines.** We compare our approach against several state-of-the-art baselines that use LLMs to generate robot control code, covering different paradigms of V&V, reasoning, and replanning. **Code as Policies (CaP)** [2] generates reusable control code from task instructions via LLM. **CaP w/ Lemur** [49] integrates SMT verification into the code generation pipeline. **CaP w/ CodeSift** [5] improves the reliability of LLM-generated code through multi-stage syntactic and semantic validation, without relying on reference code or actual execution. **LLM-Planner** [11] introduces an execution-aware replanning framework, replans after failure using new observations from the environment. **AutoGen** [50] extends LLM-Planner by enabling multi-agent collaborative reasoning.

**NESYRO implementation.** We employ `GPT-4o-mini` [51] for code generation and feedback generation. Additionally, `Llama-3.2-3B` [52] is used to compute the CSC. The decoding temperature is fixed at 0.0 for all generation steps. For the verification phase (i), the Z3 SMT solver [20] is employed as the symbolic verification tool, while for the validation phase (ii), the Fast Downward planner [53] is used as the symbolic validation tool.

Detailed descriptions of the experimental settings are provided in the Appendix B and C.

## 4.2 Main Result

**Task performance on RLBench.** Table 2 reports the performance of robot control code across all task types in dynamic, partially observable scenarios. For each task, we consider four levels

Table 3: Task performance across task types in the real-world under partial observability (*High and Low Incompleteness* averaged). NᴇSʏRᴏ-Complete reports results under *Complete*.

| Real-World | CaP | | | CaP w/ CodeSift | | | NeSyRo | | | NeSyRo-Complete | | |
|---|---|---|---|---|---|---|---|---|---|---|---|---|
| Task Type | SR (↑) | GC (↑) | IA (↓) | SR (↑) | GC (↑) | IA (↓) | SR (↑) | GC (↑) | IA (↓) | SR (↑) | GC (↑) | IA (↓) |
| Object Relocation | $7.5_{\pm3.5}$ | $11.3_{\pm1.8}$ | 19 | $12.5_{\pm3.5}$ | $19.4_{\pm4.4}$ | 4 | $82.5_{\pm3.5}$ | $83.8_{\pm3.5}$ | 2 | $85.0_{\pm7.1}$ | $90.0_{\pm3.5}$ | 2 |
| Object Interaction | $30.0_{\pm7.1}$ | $37.5_{\pm7.1}$ | 12 | $20.0_{\pm7.1}$ | $24.4_{\pm9.7}$ | 4 | $75.0_{\pm14.1}$ | $77.5_{\pm17.7}$ | 0 | $90.0_{\pm14.1}$ | $90.0_{\pm14.1}$ | 0 |
| Auxiliary Manipulation | $0.0_{\pm0.0}$ | $0.0_{\pm0.0}$ | 4 | $2.5_{\pm3.5}$ | $8.3_{\pm4.7}$ | 5 | $20.0_{\pm0.0}$ | $20.0_{\pm0.0}$ | 2 | $20.0_{\pm14.1}$ | $20.0_{\pm14.1}$ | 2 |
| Long-Horizon | $5.0_{\pm0.0}$ | $14.2_{\pm3.5}$ | 18 | $7.5_{\pm10.6}$ | $13.1_{\pm7.4}$ | 16 | $52.5_{\pm3.5}$ | $54.2_{\pm3.5}$ | 3 | $60.0_{\pm0.0}$ | $65.8_{\pm8.3}$ | 2 |
| Total | $10.6_{\pm0.9}$ | $15.7_{\pm0.4}$ | 53 | $10.6_{\pm4.4}$ | $16.3_{\pm4.4}$ | 29 | $57.5_{\pm3.5}$ | $58.9_{\pm4.4}$ | 7 | $68.8_{\pm5.3}$ | $71.5_{\pm6.5}$ | 6 |

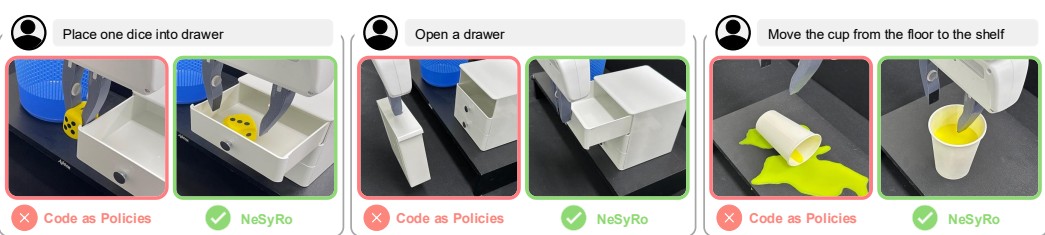

Figure 3: Representative failure scenarios under partial observability across real-world tasks

of observability incompleteness, requiring each method to recover missing observations. NᴇSʏRᴏ consistently outperforms baselines (AutoGen and CaP w/ CodeSift) by 26.3% in SR and 24.3% in GC across all levels of observability incompleteness.

CaP w/ Lemur outperforms the base CaP model, indicating that pre-execution verification improves robustness even without explicit safe probe. LLM-based replanning methods, such as AutoGen and LLM-Planner, perform well compared to V&V-based approaches, such as CaP w/ CodeSift, under the *Low Incompleteness* condition, but their performance degrades significantly as uncertainty increases and critical observations are missing. In contrast, NᴇSʏRᴏ sustains strong results across every observability regime and task type. It accurately detects missing observations and performs safe probes, avoiding irreversible actions. Under the *Complete* condition, all methods achieve high performance since no additional probe is required. As observability grows more incomplete, baseline performance drops sharply. Meanwhile, NᴇSʏRᴏ detects the missing observations, explores safely, and maintains performance close to that of the Complete setting. This tendency becomes even more pronounced in long-horizon tasks, where extensive probe is required and bridging observation gaps becomes especially difficult.

**Task performance on Real-World.** Table 3 reports the real-world evaluation results under partial observability. The NᴇSʏRᴏ-Complete configuration serves as an upper bound, representing the ideal performance achievable when all relevant observations are fully available. Across all task categories, NᴇSʏRᴏ consistently achieves the highest SR and GC, outperforming existing baselines such as CaP and CaP w/ CodeSift. On average, NᴇSʏRᴏ improves SR by an average of 47.0% and GC by 42.6% compared to CaP w/ CodeSift, while simultaneously reducing IA from 29 to 7.

In particular, for *Object Relocation* and *Object Interaction*, baseline methods achieve less than 30% SR, primarily due to their failure to account for missing observations. In contrast, NᴇSʏRᴏ exceeds 75% SR by recovering these observations. Performance on *Auxiliary Manipulation* remains low across all models, including NᴇSʏRᴏ-Complete, primarily due to failure in pressing the light switch with sufficient accuracy. Nevertheless, NᴇSʏRᴏ achieves the same SR and GC as its Complete variant, indicating upper-bound performance despite the physical difficulty. This highlights the challenge as one of execution, rather than perceptual or reasoning limitations. On *Long-Horizon* tasks, baseline methods remain below 10% SR due to observation gaps and planning complexity. Nonetheless, NᴇSʏRᴏ reaches 52.5% SR, approaching the 60.0% score of the *Complete* setting. This shows that our framework bridges long-range dependencies through safe probing. These results confirm that NᴇSʏRᴏ enables robust real-world execution through safe recovery of missing observations, even under severe observability incompleteness.

Figure 3 illustrates failure modes associated with high IA scores. Among many failure cases, we present three representative scenarios where CaP fails due to irreversible actions, while NᴇSʏRᴏ completes the tasks by recovering missing observations through safe probe. This selection illustrates that NᴇSʏRᴏ not only improves success rates but also ensures safe execution grounded in safe probe.

## 4.3 Ablation

Table 4: Code generation performance across LLMs on long-horizon tasks under partial observability in RLBench

| RLBench | CaP | | NeSyRo | | NeSyRo-Complete | |
|---|---|---|---|---|---|---|
| LLM Type | SR | GC | SR | GC | SR | GC |
| GPT-4o-mini | $10.0_{\pm0.0}$ | $20.2_{\pm3.3}$ | $45.0_{\pm0.0}$ | $61.7_{\pm0.3}$ | $65.0_{\pm7.1}$ | $73.7_{\pm8.3}$ |
| o4-mini | $12.5_{\pm3.5}$ | $22.0_{\pm1.4}$ | $42.5_{\pm3.5}$ | $54.9_{\pm2.2}$ | $50.0_{\pm14.1}$ | $51.7_{\pm11.8}$ |
| GPT-4.1 | $32.5_{\pm3.5}$ | $59.6_{\pm4.8}$ | $50.0_{\pm7.1}$ | $70.9_{\pm1.8}$ | $75.0_{\pm7.1}$ | $78.1_{\pm2.7}$ |
| o3 | $45.0_{\pm0.0}$ | $64.0_{\pm1.9}$ | $75.0_{\pm7.1}$ | $87.6_{\pm3.0}$ | $85.0_{\pm7.1}$ | $93.8_{\pm0.7}$ |

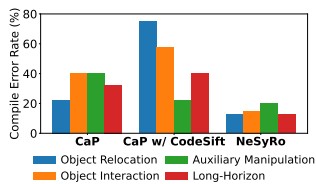

Figure 4: Real-world compile error rate over all task types

**Effect of neuro-symbolic code verification** Table 4 reports performance across different LLMs used for code generation, evaluated on long-horizon tasks and averaged over *High* and *Low Incompleteness* conditions. Stronger LLMs improve performance. On average, NESYRO improves SR by 28.1% and GC by 27.3% over CaP across all LLMs. The SR difference between NESYRO and the upper-bound NESYRO-Complete remains at 15.6% on average, indicating that our framework consistently approaches the optimal performance achievable under complete observations. This shows that NESYRO performs consistently across LLMs. Figure 4 reports real-world task failures due to compile errors across task types. NESYRO consistently exhibits the lowest compile error rate, highlighting the robustness of its verification and validation pipeline. In contrast, CaP w/ CodeSift incurs more compile errors than the base CaP under partial observability, primarily due to hallucinated evaluations by the LLM in the absence of grounded feedback. NESYRO addresses this issue through environment-aware validation, enabling reliable execution even with incomplete observations.

Table 5: Comparison of performance on long-horizon RLBench tasks when without LC or CSC from NESYRO

Table 6: Effect of LLM parameter scale on CSC in RLBench

| RLBench | NeSyRo *w/o* LC | | NeSyRo *w/o* CSC | | NeSyRo | |
|---|---|---|---|---|---|---|
| Task Type | SR | GC | SR | GC | SR | GC |
| Object Relocation | $50.0_{\pm8.2}$ | $60.0_{\pm4.1}$ | $35.0_{\pm5.8}$ | $51.3_{\pm4.8}$ | $67.5_{\pm10.6}$ | $73.8_{\pm5.3}$ |
| Object Interaction | $45.0_{\pm7.1}$ | $56.3_{\pm5.3}$ | $55.0_{\pm7.1}$ | $62.5_{\pm3.5}$ | $70.0_{\pm7.1}$ | $75.0_{\pm3.5}$ |
| Auxiliary Manipulation | $57.5_{\pm3.5}$ | $65.4_{\pm4.1}$ | $35.0_{\pm7.1}$ | $39.2_{\pm3.5}$ | $65.0_{\pm7.1}$ | $77.5_{\pm7.1}$ |
| Long-Horizon | $25.0_{\pm0.0}$ | $36.0_{\pm6.3}$ | $32.5_{\pm3.5}$ | $52.9_{\pm8.4}$ | $45.0_{\pm0.0}$ | $61.7_{\pm0.3}$ |
| Total | $44.3_{\pm2.0}$ | $54.2_{\pm0.1}$ | $37.1_{\pm4.0}$ | $49.9_{\pm4.4}$ | $61.9_{\pm6.2}$ | $72.0_{\pm3.9}$ |

| RLBench | Long-Horizon Tasks | |
|---|---|---|
| LLM Type | SR | GC |
| Llama-3.2-1B | $32.5_{\pm3.5}$ | $55.4_{\pm2.7}$ |
| Llama-3.2-3B | $45.0_{\pm0.0}$ | $61.7_{\pm0.3}$ |
| Llama-3.1-8B | $45.0_{\pm7.1}$ | $57.7_{\pm0.3}$ |
| Qwen3-30B-A3B | $45.0_{\pm0.0}$ | $64.9_{\pm5.1}$ |

**Effect of neuro-symbolic code validation** Tables 5 and 6 report results on long-horizon tasks, averaged over *High* and *Low Incompleteness* conditions. Table 5 compares performance when removing either LC (*w/o LC*) or CSC (*w/o CSC*) from our neuro-symbolic code validation phase. The results indicate that both LC and CSC contribute comparably to overall performance across task types, with an average SR drop of 21.2% and GC drop of 20.0% when either component is removed. This validates the importance of the parallel neuro-symbolic structure in reliably guiding execution under uncertainty. To analyze CSC robustness, Table 6 examines the effect of the LLM parameter scale used in computing the CSC. While smaller models (e.g., Llama-3.2-1B) show degraded performance, models with 3B parameters or more achieve the same SR and differ in GC by less than 7.2%. These results suggest that CSC computation is robust to LLM scaling beyond a moderate threshold, allowing flexible deployment depending on available compute resources.

## 4.4 Analysis

**Real-world analysis.** Figure 5 showcases how NESYRO addresses real-world uncertainty in a partially observable setting involving a dark room and unknown drawer states. Initially, the LLM generates an ungrounded policy code without access to key observations, such as whether the drawers are locked, what objects are inside, or where to safely place the dice. As each skill is validated, NESYRO computes NeSyConf. If the confidence falls below a threshold, the system initiates a targeted safe probe, such as checking individual drawers, to acquire missing observations. These safe probes trigger policy code refinement by updating parameters (e.g., selecting a different drawer) or regenerating the policy for the skill. This iterative validation process adapts the skill sequence and produces a fully grounded policy code that completes the task without causing irreversible failures. A full execution sequence and extended analysis of Figure 5 are provided in Appendix D.

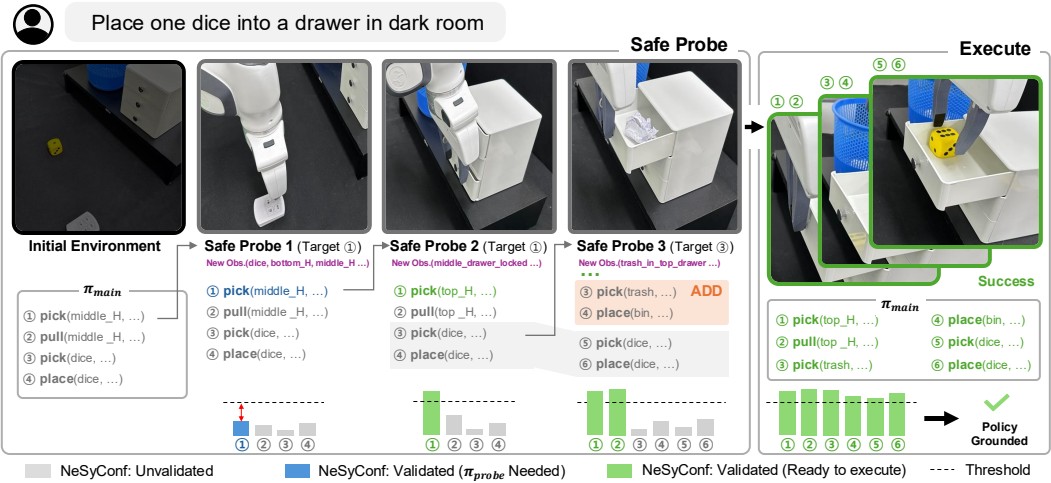

Figure 5: Real-world example of safe probe and policy code refinement in partially observable setting

## 5 Conclusion

In this work, we presented NESYRO, a neuro-symbolic framework that integrates *Neuro-symbolic Code Verification* and *Neuro-Symbolic Code Validation* to generate reliable robot control code under dynamic, partially observable settings. The framework operates through a recursive process alternating between symbolic verification and interactive validation, ensuring that each skill is both logically consistent and environmentally grounded. By incorporating neuro-symbolic confidence estimation that combines commonsense and logic-based reasoning, NESYRO enables exploratory yet safe interactions and adaptive code refinement under uncertainty. Extensive evaluation in simulation and real-world environments demonstrates the strong performance of NESYRO across diverse tasks.

**Limitation and future work.** While the NeSyConf formulation is designed to allow LC and CSC to complement each other, similar to how SayCan [1] integrates LLM scores with affordance functions, the current implementation of NESYRO employs a binary LC and predefined domain knowledge, which limits its generality in real-world applications. Future work will address this limitation by incorporating probabilistic and temporal reasoning, such as probabilistic PDDL [54]. We also plan to relax these assumptions and explore the framework's applicability to more diverse and dynamic domains by extending validation to skills that are not explicitly defined in the domain knowledge.

**Ethical concern.** LLM-based robot control may lead to unsafe behavior when interacting with hazardous tools (e.g., knives, scissors). To mitigate such risks, we incorporate explicit safety checks and enforce transparent safeguard mechanisms that verify tool affordances and action preconditions before execution, ensuring safe and interpretable operation.

## Acknowledgement

This work was supported by Institute of Information & communications Technology Planning & Evaluation (IITP) grant funded by the Korea government (MSIT), (RS-2022-II220043 (2022-0-00043), Adaptive Personality for Intelligent Agents, RS-2022-II221045 (2022-0-01045), Self-directed multi-modal Intelligence for solving unknown, open domain problems, RS-2025-02218768, Accelerated Insight Reasoning via Continual Learning, RS-2025-25442569, AI Star Fellowship Support Program (Sungkyunkwan Univ.), and RS-2019-II190421, Artificial Intelligence Graduate School Program (Sungkyunkwan University)), the National Research Foundation of Korea (NRF) grant funded by the Korea government (MSIT) (No. RS-2023-00213118), IITP-ITRC (Information Technology Research Center) grant funded by the Korea government (MIST) (IITP-2025-RS-2024-00437633, 10%), IITP-ICT Creative Consilience Program grant funded by the Korea government (MSIT) (IITP-2025-RS-2020-II201821, 10%), and by Samsung Electronics.

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

# A   Details of the NESYRO Framework

NESYRO enables exploration through a recursive composition of Neuro-symbolic Code Verification and Neuro-symbolic Code Validation. A policy code produced by the LLM is first subjected to explicit symbolic verification, which statically checks logical consistency against the task specification. The verified policy code then enters interactive validation, where each skill is evaluated in neuro-symbolic manner; if an unmet precondition is detected, the system synthesizes exploratory safe probe policy code to gather the missing observations. Every probe is fed back through the same verification and validation (V&V) cycle, producing a policy tree whose nodes are recursively grounded until all skills achieve a satisfactory Neuro-symbolic Confidence Score. This recursive V&V framework guarantees that the policy code is both executable and environmentally grounded, even under dynamic, partially observable conditions.

## A.1   NESYRO Algorithm

We provide the full pseudocode for our neuro-symbolic task execution pipeline in Algorithm 1 and Algorithm 2. Below, we briefly describe the functional roles of each procedure and its interaction within the recursive planning framework.

---

**Algorithm 1** Task Execution Pipeline

---

**Agent:**
  env — environment interface     $\mathcal{D}$ — domain knowledge
  $\mathcal{E}_{\text{demo}}$ — demonstration set     $\epsilon$ — confidence threshold
  $a$ — primitive action     $o_{\leq t}$ — observation history up to $t$
  $\mathcal{A}$ — action set

**Returns:**
  $\tau$ — executed trajectory     $\pi_{\text{main}}$ — policy code

**procedure** RUNTASK$(\text{env}, \mathcal{D}, \mathcal{E}_{\text{demo}}, \epsilon)$
  $(o_{\leq 0}, g) \leftarrow \text{env.reset}()$
  $\tau \leftarrow ()$                 ▷ Initialize empty trajectory
  $(\pi_{\text{main}}, o_{\leq t}, \text{env}, \tau) \leftarrow \text{NESYRO}(g, o_{\leq 0}, \mathcal{D}, \mathcal{E}_{\text{demo}}, \epsilon, 0, \text{env}, \tau)$
  $(\tau_{\text{exe}}, o_{\leq t+1}, \text{env}) \leftarrow \text{EXE}(\pi_{\text{main}}, o_{\leq t}, \text{env})$
  $\tau \leftarrow \tau \cup \tau_{\text{exe}}$
  **return** $(\tau, o_{\leq t+1}, \pi_{\text{main}})$
**end procedure**

**procedure** EXE$(\pi, o_{\leq t}, \text{env})$
  $\tau_{\text{exe}} \leftarrow [\,]$
  **for each** $f$ **in** $\pi$ **do**
   $\mathcal{A} \leftarrow \text{EXPANDSKILL}(f, o_{\leq t})$
   **for each** $a$ **in** $\mathcal{A}$ **do**
    $o_{\text{next}} \leftarrow \text{env.step}(a)$
    $\tau_{\text{exe}} \leftarrow \tau_{\text{exe}} \cup (a, o_{\text{next}})$
    $o_{\leq t} \leftarrow o_{\leq t} \cup o_{\text{next}}$
   **end for**
  **end for**
  **return** $(\tau_{\text{exe}}, o_{\leq t}, \text{env})$
**end procedure**

---

**RunTask and execution loop.** The RUNTASK procedure (Algorithm 1) initializes the environment and launches the neuro-symbolic reasoning process. It first resets the environment to obtain the initial observation $o_{\leq 0}$ and instruction $g$, and initializes an empty trajectory $\tau$. The main grounding routine NESYRO is then called to synthesize a grounded policy code $\pi_{\text{main}}$ based on the instruction and current context. Once obtained, the policy is executed via EXE, which expands symbolic skills into primitive actions and steps through the environment. The complete trajectory $\tau$ and updated observation history $o_{\leq t}$ are returned. The EXE procedure handles the execution of symbolic skills. For

each skill $f$ in $\pi$, it calls EXPANDSKILL to retrieve the sequence of corresponding low-level actions. These are executed sequentially in the environment, and the resulting observations are appended to both the trajectory and the observation history.

---

**Algorithm 2** Recursive Neuro-symbolic Verification & Validation

---

   **procedure** NESYRO$(g, o_{\leq t}, \mathcal{D}, \mathcal{E}_{\text{demo}}, \epsilon, k, \text{env}, \tau)$
      $(\mathcal{T}_{\text{spec}}, \pi_{\text{main}}) \leftarrow$ NEURO_SYMBOLIC_VERIFICATION$(g, o_{\leq t}, \mathcal{D}, k)$
      $(\pi_{\text{main}}, o_{\leq t}, \text{env}, \tau) \leftarrow$
             NEURO_SYMBOLIC_VALIDATION$(g, o_{\leq t}, \mathcal{D}, \mathcal{E}_{\text{demo}}, \pi_{\text{main}}, \epsilon, k, \text{env}, \tau)$
      **return** $(\pi_{\text{main}}, o_{\leq t}, \text{env}, \tau)$                         ▷ grounding policy code $\pi_{\text{main}}$
   **end procedure**

   **procedure** NEURO_SYMBOLIC_VERIFICATION$(g, o_{\leq t}, \mathcal{D}, k)$
      $(\mathcal{T}_{\text{spec}}, \pi_{\text{main}}) \leftarrow \Phi_{\text{veri}}(o_{\leq t}, g, l_{\text{cot}}, \mathcal{D}, k)$
      **while** $\Psi_{\text{veri}}(\mathcal{T}_{\text{spec}}, \pi_{\text{main}}) = \mathsf{fail}$ **do**
         $\mathcal{F}_{\text{veri}} \leftarrow \Psi_{\text{veri}}(\mathcal{T}_{\text{spec}}, \pi_{\text{main}})$
         $(\mathcal{T}_{\text{spec}}, \pi_{\text{main}}) \leftarrow \Phi_{\text{veri}}(o_{\leq t}, g, l_{\text{cot}}, \mathcal{D}, \pi_{\text{main}}, \mathcal{F}_{\text{veri}}, k)$
      **end while**
      **return** $(\mathcal{T}_{\text{spec}}, \pi_{\text{main}})$
   **end procedure**

   **procedure** NEURO_SYMBOLIC_VALIDATION$(g, o_{\leq t}, \mathcal{D}, \mathcal{E}_{\text{demo}}, \pi_{\text{main}}, \epsilon, k, \text{env}, \tau)$
      $n \leftarrow k$
      **while** $n < |\pi_{\text{main}}|$ **do**
         $f_n \leftarrow \pi_{\text{main}}[n]$
         $\text{CSC} \leftarrow \Phi_{\text{vali}}(\mathcal{D}, \mathcal{E}_{\text{demo}}, o_{\leq t}, g, f_n)$
         $\text{LC} \leftarrow \Psi_{\text{vali}}(\mathcal{D}, o_{\leq t}, g, f_n)$
         $\text{NeSyConf} \leftarrow \text{CSC} \times \text{LC}$
         **if** $\text{NeSyConf} < \epsilon$ **then**
            $g_{\text{probe}} \leftarrow$ MAKEPROBEGOAL$(f_n, \mathcal{F}_{\text{csc}}, \mathcal{F}_{\text{lc}})$
            $(\pi_{\text{probe}}, o_{\leq t}, \text{env}, \tau) \leftarrow$ NESYRO$(g_{\text{probe}}, o_{\leq t}, \mathcal{D}, \mathcal{E}_{\text{demo}}, \epsilon, 0, \text{env}, \tau)$       ▷ recursive
            $(\tau_{\text{exe}}, o_{\leq t+1}, \text{env}) \leftarrow$ EXE$(\pi_{\text{probe}}, o_{\leq t}, \text{env})$
            $\tau \leftarrow \tau \cup \tau_{\text{exe}}$
            $(\mathcal{T}_{\text{spec}}, \pi_{\text{main}}) \leftarrow$ NEURO_SYMBOLIC_VERIFICATION$(g, o_{\leq t+1}, \mathcal{D}, n)$
         **else**
            $n \leftarrow n + 1$
         **end if**
      **end while**
      **return** $(\pi_{\text{main}}, o_{\leq t+\alpha}, \text{env}, \tau)$     ▷ $\alpha$: number of recursive $\pi_{\text{probe}}$ executed during validation.
   **end procedure**

---

**Recursive neuro-symbolic reasoning.** Algorithm 2 outlines the recursive grounding logic of NESYRO. The NESYRO procedure first invokes NEURO_SYMBOLIC_VERIFICATION to obtain a symbolic task specification $\mathcal{T}_{\text{spec}}$ and initial policy code $\pi_{\text{main}}$. Logical correctness is ensured through iterative verification using $\Phi_{\text{veri}}$ and $\Psi_{\text{veri}}$, which checks whether $\pi_{\text{main}}$ satisfies $\mathcal{T}_{\text{spec}}$. After verification, the policy is passed to NEURO_SYMBOLIC_VALIDATION for skill-wise confidence assessment. For each skill $f_n$, the framework computes neuro-symbolic confidence score (NeSyConf). If NeSyConf falls below threshold $\epsilon$, a probing goal $g_{\text{probe}}$ is generated and recursively passed into NESYRO.

To construct this probing goal $g_{\text{probe}}$, the MAKEPROBEGOAL function synthesizes a new instruction that addresses the failure feedback. Specifically, it leverages NeSyConf feedback $(\mathcal{F}_{\text{csc}}, \mathcal{F}_{\text{lc}})$ to identify missing observations. This recursive routine allows a skill that fails to exceed the confidence threshold $\epsilon$ to be refined and validated using updated observations gathered from safe probe executions. Once all skills pass validation, the final grounded policy and accumulated trajectory are returned.

# B  Environment Settings

## B.1  RLBench

We use RLBench [8] as the simulation environment for our experiments. RLBench offers a wide range of tabletop manipulation tasks and provides realistic simulations of both robot control and visual observations. All experiments are performed using a 7-DoF Franka Emika Panda robotic arm, which is supported natively by RLBench. The environment is particularly suitable for evaluating planning and interaction under partial observability, as it supports randomized object configurations and sensor data, including RGB, depth, and segmentation masks. Its compatibility with Python also allows straightforward integration with our code generation and execution framework.

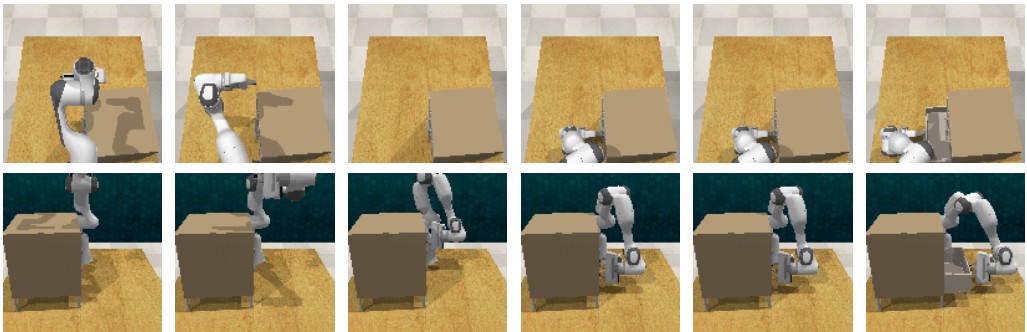

Figure 6: Example scenes illustrating the "open drawer" task in the RLBench. The top row shows the overhead view, and the bottom row shows the front view.

**Object configuration.**  Each episode initializes a workspace containing seven unique objects placed on a table. The objects include two tomatoes, a piece of trash, a bin, a three-level drawer, a desk lamp, and a switch for the desk lamp. The position of each object is randomized in every episode, introducing perceptual variability and scene diversity across tasks. Figure 6 shows an example scene from the RLBench environment used in our experiments.

**Task composition.**  We categorize the tasks into four types to enable structured evaluation, as summarized in Table 7. Each task type contains multiple language instructions, with associated probe targets indicating the source of uncertainty that must be resolved during execution.

Table 7: Task types, example of instructions, and associated probe targets in RLBench.

| Task Type | Example Instructions | Probe Target |
|---|---|---|
| Object Relocation | Move two tomatoes onto plate
Put the trash into bin | Tomato identity
Trash identity |
| Object Interaction | Open a drawer
Open two drawers | Drawer locked/unlocked state
Drawer locked/unlocked state |
| Auxiliary Manipulation | Move two tomatoes onto plate in dark room
Open drawer in dark room | Missing visual observation (requires light activation)
Missing visual observation (requires light activation) |
| Long-Horizon Tasks | Move a die into the drawer
Move dice into the drawer | Die identity, Drawer locked/unlocked state
Dice identity, Drawer locked/unlocked state |

## B.2  Real-world

**Environment setup.**  We conducted our real-world experiments using a 7-DoF Franka Emika Research 3 robotic arm mounted on a tabletop workspace. An Intel RealSense D435 RGB-D camera was positioned above the table to provide top-down RGB and depth information. This input was processed by an object detection module to identify the categories and bounding boxes of task-relevant objects. Depth measurements were used to compute 3D coordinates, which were then transformed into the robot's coordinate frame. This setup enabled accurate object localization and real-time observation grounding, providing the necessary perception for reliable execution.

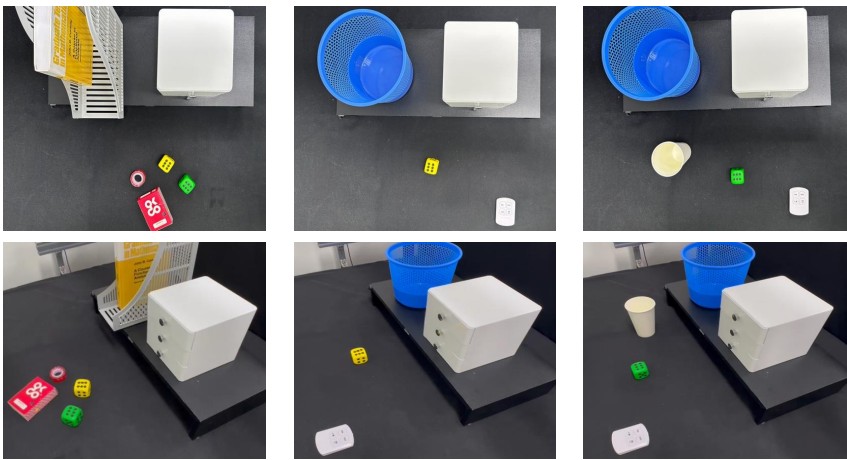

Figure 7: Example scenes from the real-world environment used in our experiments.

**Object configuration.** The real-world environment contains ten unique objects arranged on a tabletop workspace. These include two dice, two pieces of trash, a bin, a three-level drawer, a small cardboard box, a paper cup filled with liquid, a roll of tape, and a light switch. The initial positions of all objects are randomized for each trial, introducing diverse spatial configurations and observation conditions across task instances. This variability supports evaluation under partial observability and enables direct comparison with the RLBench-based simulation setup. Figure 7 shows a representative setup of the real-world environment used in our experiments.

**Task composition.** We maintain the RLBench task categorization in the real-world setup to ensure consistency and enable direct comparison. Each of the four task types corresponds to a distinct source of uncertainty and is associated with multiple language instructions and probe targets, as summarized in Table 8.

Table 8: Task types, example of instructions, and associated probe targets in real-world.

| Task Type | Example Instructions | Probe Target |
|---|---|---|
| Object Relocation | Place a die into drawer
Move dice into drawer | Die identity
Dice identity |
| Object Interaction | Open a drawer
Open two drawers | Drawer locked/unlocked state
Drawer locked/unlocked state |
| Auxiliary Manipulation | Place a die into drawer in dark room
Open drawer in dark room | Missing visual observation (requires light activation)
Missing visual observation (requires light activation) |
| Long-Horizon Tasks | Place a die into drawer
Move a die into drawer | Die identity, Drawer locked/unlocked state
Die identity, Drawer locked/unlocked, empty/occupied state |

**Low-Level Control.** For motion planning and control in the real-world environment, we employed MoveIt [55], an open-source motion planning framework widely used for robotic manipulation. Once the target object positions were obtained from the perception pipeline, we invoked parameterized skill primitives such as `pick`, `place`, and `open`, which are designed to operate over arbitrary object poses. Each skill was instantiated using the transformed 3D coordinates of the corresponding object and passed to the planner as goal constraints. Trajectory optimization was handled by MoveIt's built-in planners, which computed collision-free joint-space paths that respect the robot's kinematic limits and workspace constraints. The resulting trajectories were executed using the robot's internal controller through ROS. Although continuous force control was not used for the gripper, we implemented discrete grasping strategies based on object geometry and semantic role (e.g., trash, dice). This ensured consistent and safe execution across a variety of physical configurations.

Unlike the RLBench setting, where code execution is simulated through parameterized low-level APIs, our real-world system closes the loop by grounding skill calls with real sensor observations and executing planned trajectories on physical hardware. This setup allows us to evaluate the reliability of the proposed planning framework under real-world uncertainties.

# C Experiment Details

## C.1 Compute Resources

Most experiments were conducted on a local machine with an Intel(R) Core(TM) i7-9700KF CPU and an NVIDIA GeForce RTX 4080 GPU (16GB VRAM). Each task instance used a single GPU, and RLBench simulation was executed with up to 32GB of system memory. Symbolic verification and PDDL planning were run on the CPU. For experiments using the larger language models listed in Table 6 in main paper, such as Llama-3.1-8B and Qwen3-30B-A3B, we used a cloud-based CUDA cluster with GPUs equipped with approximately 82GB of VRAM. All OpenAI models, including GPT-4o and GPT-4.1, were accessed via the OpenAI API.

## C.2 NeSyRo Implementation

**LLM usage overview.** Our framework utilizes LLM as core reasoning engines in three tightly integrated components of the code generation and validation pipeline:

- **Code Generation ($\Phi_{\textbf{veri}}$):** Given the instruction $g$ and the current observation history $o_{\leq t}$, the verification LLM $\Phi_{\text{veri}}$ performs chain-of-thought reasoning to produce an intermediate symbolic task specification $\mathcal{T}_{\text{spec}}$ and corresponding Python policy code $\pi_{\text{main}}$. If the code fails symbolic verification via $\Psi_{\text{veri}}$, structured feedback $\mathcal{F}_{\text{veri}}$ is returned and used by the LLM to iteratively revise the code. Only the unvalidated portion of $\pi_{\text{main}}$ is regenerated at each step, preserving previously verified components.

- **CSC Computation ($\Phi_{\textbf{vali}}$):** For each skill $f_n$ in the primary policy $\pi_{\text{main}}$, the validation LLM $\Phi_{\text{vali}}$ computes $\text{CSC}_{f_n}$ that estimates the likelihood of successful execution under the current observation $o_{\leq t}$ and instruction $g$. The LLM is prompted with the code for $f_n$, domain knowledge $\mathcal{D}$, retrieved single-skill demonstrations $\mathcal{E}_{\text{demo}}$, and task context. Token-level probabilities are aggregated and transformed into a negative log-likelihood score. This value is normalized to produce a scalar confidence score $\text{CSC}_{f_n}$ used for validation. Before normalization, $\text{CSC}_{f_n}$ ranges over $[0, \infty)$; after normalization, it is scaled to the interval $[0, 1]$.

- **CSC Feedback Generation ($\mathcal{F}_{\textbf{csc}}$):** If $\text{NeSyConf}_{f_n} < \epsilon$, the skill $f_n$ is considered to require a safe probe. In such cases, CSC feedback $\mathcal{F}_{\text{csc}}$ is constructed based on $f_n$, the failure context, current observation $o_{\leq t}$, instruction $g$, and single-skill demonstrations $\mathcal{E}_{\text{demo}}$. This feedback is then used to prompt the LLM to generate a safe probe policy code $\pi_{\text{probe}}$. The resulting policy code is recursively verified and validated through the NeSyRo pipeline before execution.

**Example of prompt.** Below are the representative prompts used in each stage of our framework: generating executable robot code, computing CSC for each skill, and generating feedback when the NeSyConf falls below a threshold.

---

**Code Generator Prompt**

**Role:** You are an AI assistant tasked with generating executable Python code that controls a robot in a simulated environment.

**Task:** Complete the executable code using the provided inputs and by implementing any missing skills. The goal is to ensure the robot can achieve the specified objective by executing a sequence of actions (plan).

**Input Details:**

- **Domain PDDL:** Describes the available actions and predicates in the environment. It includes information about action parameters, preconditions, and effects. This provides the symbolic action space for the planner.
  `Provided domain PDDL: {{domain_pddl}}`

- **Observation (Initial State Description):** Represents the initial state of the environment in PDDL format, including locations of objects, robot position, and other relevant state descriptions.
  `Provided observation: {{observation}}`

---

- **Goal (Natural Language Description):** The goal specifies what the robot must accomplish in plain language.
  `Provided goal: {{goal}}`

- **Specification (Code Generation Guidelines):** Provides strict rules and constraints that the generated code must follow.
  `Provided specification: {{spec}}`

- **Skill Code (Python Implementations of Actions):** A set of predefined Python functions that implement low-level skills (e.g., move, pick, place).
  `Provided skill code: {{skill_code}}`

- **Executable Code Skeleton:** A partially completed Python file containing environment setup and control flow scaffolding.
  `Provided skeleton: {{skeleton_code}}`

- **Available Skill Names:** A list of all valid skill function names that may be called in the generated code.
  `Provided skills: {{available_skills}}`

- **Object List:** A list of object names that exist in the environment.
  `Provided object list: {{object_list_position}}`

- **Feedback:** Corrections or issues identified from the previous code generation.
  `Feedback: {{feedback}}`
  `Previous code: {{prev_code}}`

- **Exploration Knowledge:** Optional knowledge to infer or explore missing observations.
  `Exploration knowledge: {{exploration_knowledge}}`

- **Frozen Code:** Code that must remain unchanged. New code must follow this segment.
  `[Frozen Code Start] {{frozen_code_part}} [Frozen Code End]`

**Implementation Requirements:**

- Use only the predefined skills (e.g., `move`, `pick`, `place`) from `skill_code`; do not define new functions.

- Complete the provided skeleton by inserting plan logic that achieves the specified goal.

- Preserve all existing imports and `[Frozen Code]` segments.

- Output should be plain text only — do not use code blocks.

- Handle errors gracefully during skill execution (e.g., invalid arguments or missing objects).

- Use the following external modules as provided:

  - `env`: Environment setup and shutdown.
  - `skill_code`: Contains all callable action implementations.
  - `video`: Tools for simulation recording.
  - `object_positions`: For retrieving object location information.

## CSC Computation Prompt

**Role:** You are an AI assistant responsible for estimating the likelihood that a specific robotic skill will succeed in the current environment. Your evaluation will be used to compute a token-level log-probability–based common sense confidence, rather than to generate output.

**Input Details:**

- **Instruction:**
  ```
  {instruction}
  ```
- **Demonstrations:**
  ```
  Demo 1: {demo_1}
  Demo 2: {demo_2}
  ...
  ```
- **Observation:**
  ```
  {observation}
  ```
- **Object List:**
  ```
  {object_list}
  ```
- **Skill Code:**
  ```
  {skill_code}
  ```

## CSC Feedback Prompt

**Role:** You are an AI assistant responsible for analyzing why a robotic skill code is likely to fail and for generating feedback to guide its refinement.

**Task:** Based on the given context and the Neuro-Symbolic Confidence Score (NeSyConf) being below threshold, provide structured feedback to help revise or improve the given skill code.

**Input Details:**

- **Demonstrations:**
  ```
  Demo 1: {demo_1}
  Demo 2: {demo_2}
  ...
  ```
- **Observation:**
  ```
  {observation}
  ```
- **Skill Code:**
  ```
  {skill_code}
  ```
- **Confidence Score**
  ```
  Current Confidence Score: {NeSyConf}, Threshold: {threshold}
  ```
- **Instruction:**
  ```
  {instruction}
  ```
- **Object List:**
  ```
  {object_list}
  ```

**Format:**

- ```
  1. Problem Identification
  ```
- ```
  2. Justification (Why is it a problem?)
  ```
- ```
  3. Proposed Solutions (High-level ideas)
  ```
- ```
  4. Additional Notes (Optional)
  ```

**Instruction to Model:** Focus on issues such as force calibration, missing objects, logical flaws, or unsafe execution. For example: "An object declared in the code is not in the actual object list." Your feedback will guide the regeneration of this skill step.

**Hyperparameter setting.** The only hyperparameter in our framework is the confidence threshold $\epsilon$ used during neuro-symbolic validation. For each skill, we perform five safe exploration probes

under varied initial conditions to estimate its execution confidence. To determine a suitable value of $\epsilon$ for a given environment, we exclude outlier trials in which the probe failed due to non-informative reasons, which could otherwise deflate confidence estimates. This ensures that $\epsilon$ reflects a realistic and actionable lower bound of confidence for successfully grounded skills.

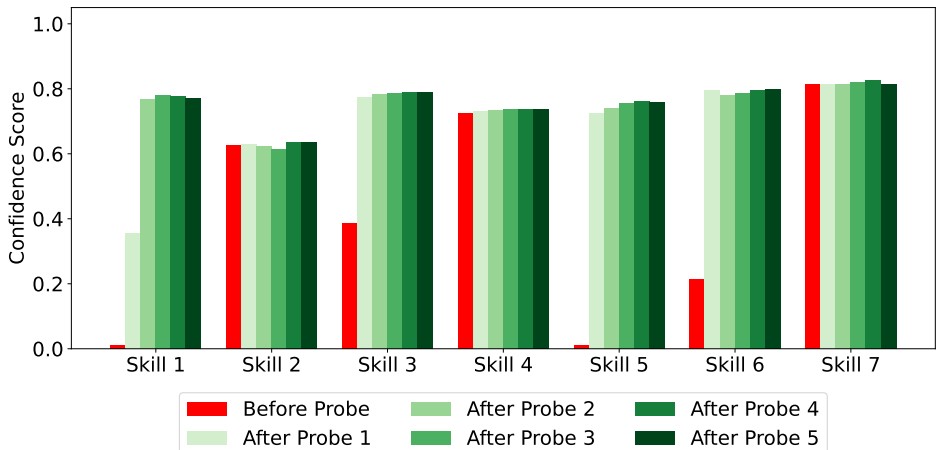

Figure 8: Confidence scores over five safe probes for each of 7 skills in a long-horizon task.

Figure 8 illustrates this process using one of the long-horizon tasks in the RLBench environment. Each of the 7 skills was probed five times, and confidence scores were recorded before and after each probe. The figure shows that while confidence increases with repeated probing, it typically saturates after a few trials, indicating convergence. Based on this observation, we compute the final confidence distribution by averaging only those probe outcomes that resulted in a successful skill grounding. We then set $\epsilon$ to the lower quartile of this filtered distribution, ensuring a conservative yet robust threshold that filters out unreliable executions while accepting skills with moderately confident grounding.

**Single-skill demonstrations format.** To support symbolic validation and LLM-based reasoning, we constructed a general-purpose demonstration library consisting of synthetic examples. These demonstrations were generated entirely via a large language model (GPT-4o) [56], given domain-level PDDL definitions and representative symbolic contexts, without requiring environment-specific execution or human annotation. Each example encodes typical task-relevant transitions across common household activities (e.g., opening a drawer, placing an object, turning on a light), and captures both successful and failure cases under varied symbolic states. In total, we synthesized approximately **500** such demonstrations spanning over **15 diverse skill types**. These examples are reused across all tasks to provide reusable prior knowledge for CSC computation and to guide safe probing decisions when symbolic grounding confidence is low.

> **Demonstration Structure**
>
> Each demonstration consists of a sequence of transitions. Each transition is represented as a dictionary with the following fields:
>
> - `initial_observation`: symbolic or structured observation before executing the action.
> - `action`: the skill function executed, such as a call to `pick(object)` or `open(drawer)`.
> - `post_observation`: symbolic or structured observation after executing the action.
> - `success`: boolean value indicating whether the action was successful.

### C.3 Baselines Implementation

**Code as Policies (CaP)** [2] serves as the foundation of our framework and is implemented by invoking the Code Generator Prompt with the frozen code length set to zero. Although a symbolic specification is also produced, this baseline does not include any verification process.

**CaP w/ Lemur** [49] extends CaP by performing verification over the generated specification. This process is conducted in the exact same manner as the Neuro-symbolic verification phase.

**CaP w/ CodeSift** [5] extends CaP by incorporating LLM-based verification and validation. In the verification stage, CodeSift performs static syntax checks using language-specific tools (`pylint` for Python, `shellcheck` for Bash) and prompts the LLM to summarize the code's functionality. This summary is then used in the validation stage to assess semantic alignment with the original task instruction. The validation consists of multiple sub-steps: semantic similarity scoring, listing all functional mismatches, and determining whether the implementation is exact. If the code fails validation, the framework automatically generates refinement feedback and prompts the LLM to revise the code accordingly. The entire process is orchestrated via a modular pipeline that yields detailed diagnostic outputs and a refined version of the code when necessary.

**LLM-Planner** [11] follows the same initial procedure as CaP by generating code from the instruction using a Code Generator Prompt. During execution in the environment, if an action fails, the planner captures the current observation and provides it as additional context to the LLM. The previously executed portion of the code is marked as frozen, and a new code segment is generated to continue the task from the failure point. As in CaP, a symbolic specification is produced, but no verification or validation is performed throughout the process.

**AutoGen** [50] adopts the same iterative replanning strategy as LLM-Planner, where code is regenerated during execution upon failure by freezing the executed portion and providing the current observation as context. The key difference is that it uses a dedicated reasoning model, specifically `o4-mini`, to enhance task understanding and decision making. This improved reasoning enables more accurate replanning. As with LLM-Planner, no explicit verification or validation is performed.

# D  Real-world Experiment Details

## D.1  Figure 1 in Main Paper Details

In this section, we provide a detailed explanation of Figure 1 in main paper. The complete execution sequence depicted in Figure 1 including all safe probes is illustrated in Figure 9. To demonstrate the reliability of NESYRO in a real-world setting, we tasked an embodied agent with the instruction "Clean up the desk" and compared NESYRO against a naive code generation approach. As depicted in Figure 1, the naive approach failed in partially observable environments, leading it to execute an irreversible action by not recognizing that the middle drawer might be locked. In contrast, NESYRO addresses this uncertainty using a safe probe pipeline to acquire the missing observations. It initially plans a safe probe to determine whether the drawers are empty. However, through its recursive validation phase, it subsequently identifies the need to observe the locked status of the drawers. Consequently, Safe Probe 1, which checks the locked status of the drawers, is executed first, as shown in Figure 9. Upon its completion, the agent adds observations confirming that the middle drawer is locked and that the top and bottom drawers are unlocked. Subsequently, Safe Probe 2, which checks whether the drawers are empty, is executed and adds observations confirming that both the top and bottom drawers are empty. With these observations acquired, the policy code is now grounded. NESYRO proceeds to successfully execute the "Clean up the desk" instruction.

## D.2  Figure 5 in Main Paper Details

This section provides a detailed explanation of Figure 5 in main paper. The complete execution sequence depicted in Figure 5 of main paper including all safe probes is illustrated in Figure 10. To further evaluate the robustness of NESYRO in a real-world setting, we implemented the instruction "Place one dice into a drawer in a dark room" which represents partially observable environments. This requires auxiliary manipulation such as turning on the light to restore visibility before executing the main task. The initial policy code $\pi_{main}$ was ungrounded, missing observations regarding drawer visibility and lock status. To resolve this, NESYRO activates its safe probe pipeline and first generates Safe Probe 1 to turn on the light, enabling the agent to perceive object locations. However, even after Safe Probe 1, the NeSyConf for the skill `pick(middle_H, ...)` remains below a threshold. In response, NESYRO generates Safe Probe 2 to check the lock status of the drawers. This probe confirms that the top and bottom drawers are unlocked, while the middle drawer is locked. Based on these observations, the code is refined by replacing the initial skills that placed one dice into the middle drawer with new skills that place it into the top drawer. As a result, the skill `pick(top_H, ...)` becomes ready to execute. Subsequently, during the validation of `pick(dice, ...)`, the agent identifies the need to check whether the drawers are empty. Consequently, NESYRO generates Safe Probe 3 to check whether the drawers are empty. This probe detects trash inside the top drawer, leading to the insertion of additional code that removes it. These added skills also undergo the same validation phase in sequence. Once all skills are marked as ready to execute, the policy code is considered grounded. NESYRO proceeds to successfully execute the given instruction.

**Instruction:** Clean up the desk

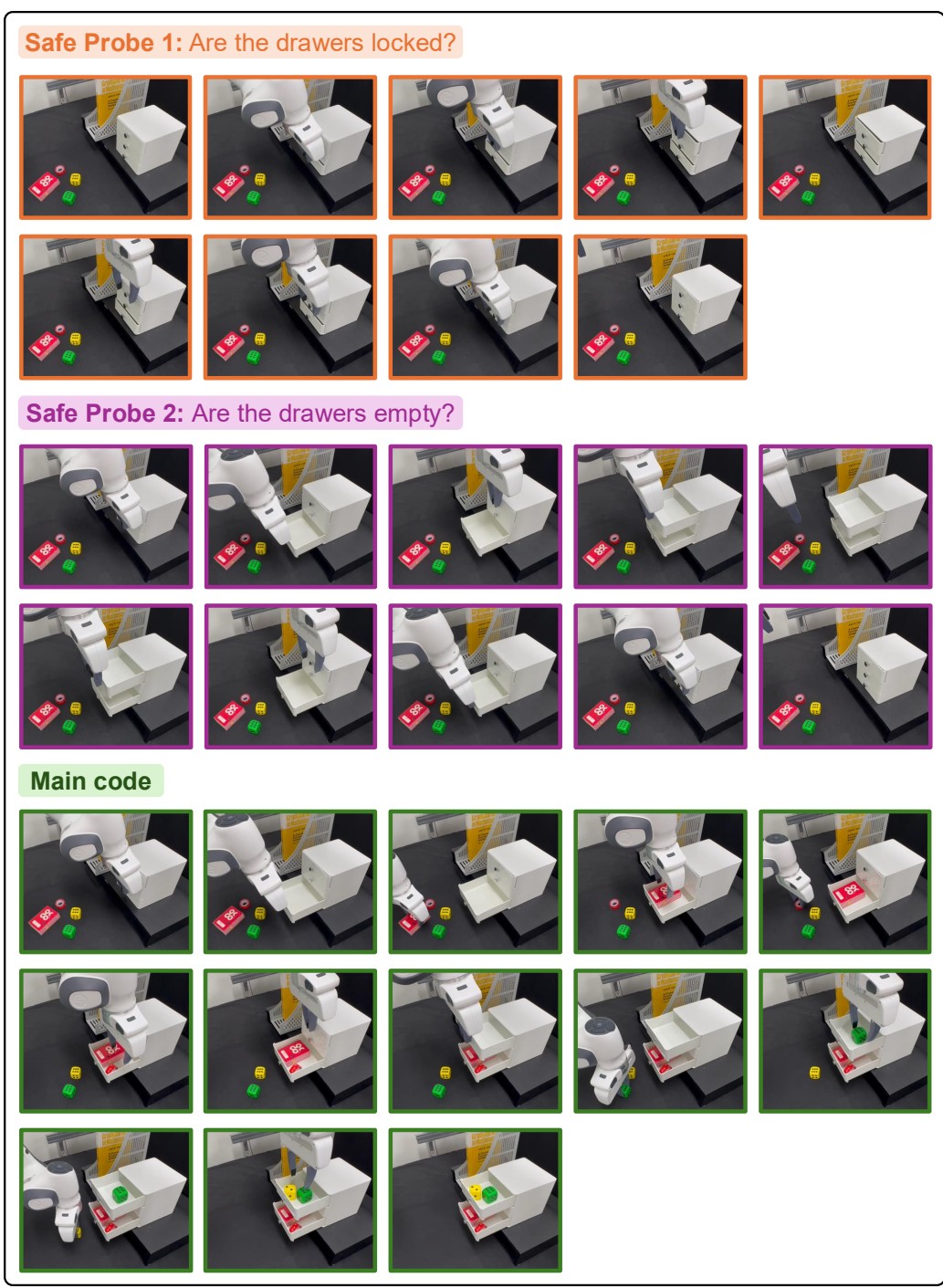

Figure 9: Real-world execution sequence of the instruction "Clean up the desk"

**Instruction:** Place one dice into a drawer in dark room

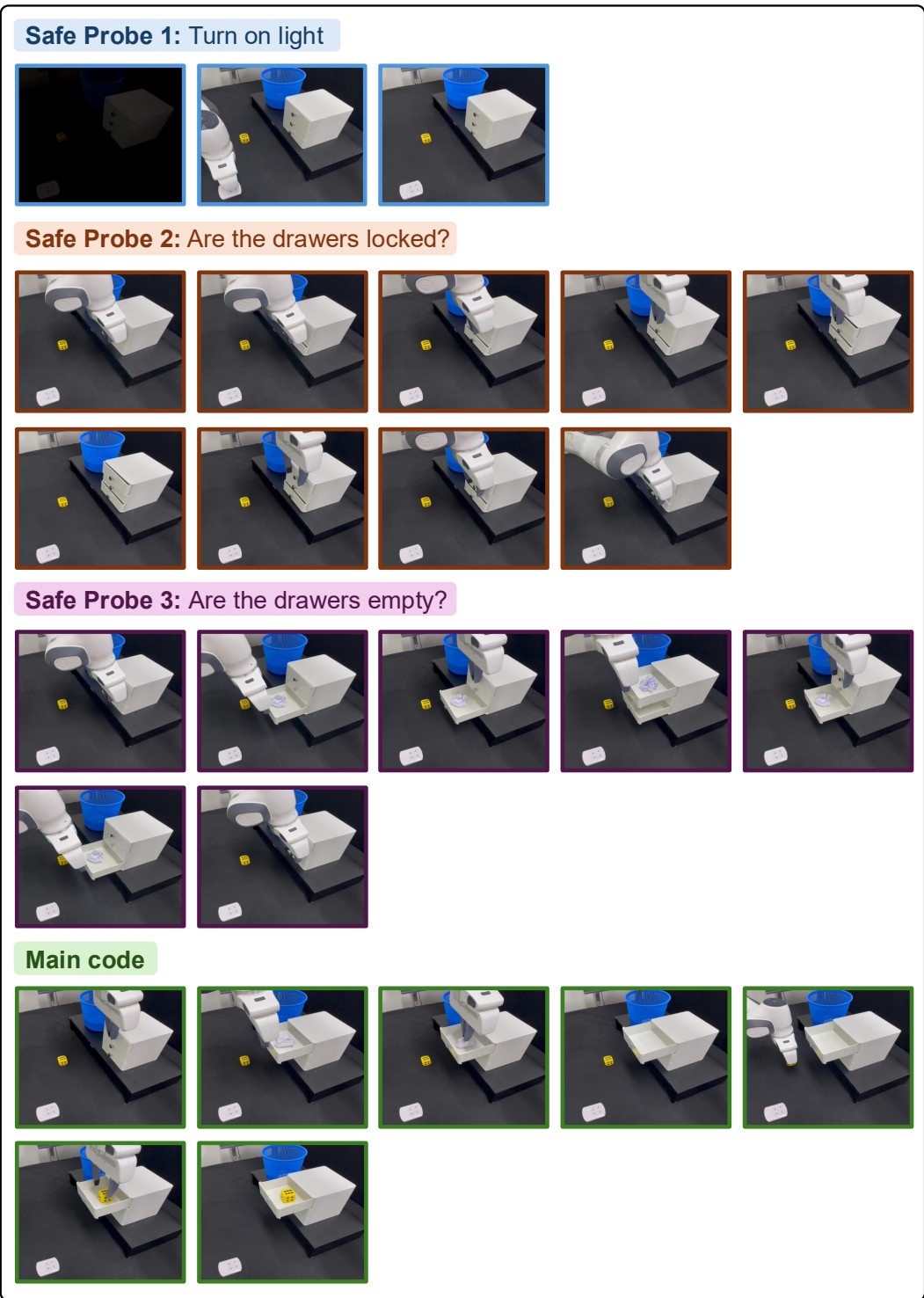

Figure 10: Real-world execution sequence of the instruction "Place one dice into a drawer in dark room"

