# OpenReview forum: "Towards Reliable Code-as-Policies: A Neuro-Symbolic Framework for Embodied Task Planning"
_NeurIPS.cc/2025/Conference — NeurIPS 2025 spotlight_

### Official Review · Reviewer_UanP · 2025-06-08

**Clarity:** 4
**Significance:** 4
**Originality:** 3
**Rating:** 5
**Confidence:** 3

**Summary:**

This paper introduces NESYRO, a neuro-symbolic framework designed to improve the reliability of LLM-generated code for robotic task planning, particularly in environments with partial observability. The core contribution is a recursive planning loop that integrates code generation with two feedback mechanisms: a symbolic verification stage that checks logical correctness, and an interactive validation stage that assesses environmental feasibility. When the framework's confidence score (NeSyConf) for a planned action is low, indicating missing information crucial for success, it autonomously generates and executes a "safe probe." This probe is an exploratory sub-policy designed to gather necessary observations from the environment (e.g., checking if a drawer is locked before trying to open it). The new information is then fed back to the LLM to refine the original plan. This process of generating, verifying, validating, and probing repeats until a high-confidence, grounded plan is produced. The authors demonstrate through extensive experiments in both the RLBench simulator and the real world that NESYRO significantly outperforms state-of-the-art baselines, improving task success rates by over 46% and ensuring high executability of actions in dynamic settings.

**Questions:**

1. The recursive validation loop is a powerful idea for ensuring robustness. However, what is the trade-off in terms of computational time and overall task completion time? Could you report on the average number of probe-refinement cycles required for the different levels of observability incompleteness and the associated latency?
2. How does the system guarantee that a probe action is indeed safe and reversible? Is this based on a manually curated set of "safe" skills, or can the symbolic verification module reason about the potential side effects of a probe?
3. The experiments are performed on tabletop manipulation tasks. How do you see the NESYRO framework scaling to more complex, longer-horizon tasks, such as mobile manipulation in a full home environment? Would the complexity of symbolic verification and the state space make the process intractable?

**Ethical Concerns:**

["NO or VERY MINOR ethics concerns only"]

**Final Justification:**

The authors have addressed my concerns, and I do not find other major flaws in this paper.

**Limitations:**

yes

**Quality:**

4

**Strengths And Weaknesses:**

### **Strengths**
1. The reliability of LLM-generated plans in partially observable, dynamic environments is a major bottleneck for real-world robotics. This paper tackles this head-on with a practical and effective solution. The focus on grounding high-level plans in low-level, verifiable actions is exactly what the field needs to move beyond constrained lab demos.
2. The NESYRO framework is technically solid. Its recursive verify-and-validate loop is effective, and the "safe probe" elegantly enables active uncertainty reduction. The hybrid confidence score (NeSyConf), combining learned commonsense and symbolic logic, thoughtfully guides the probing process.
3. The paper’s evidence is robust: tested on four task categories and observability levels in both simulation and hardware, the method outperforms strong baselines (CaP, CaP + static analysis, AutoGen) by 46.2 % and reaches 86.8 % executability in the real world. Ablations show that each module (LC, CSC) is essential.
4. By testing on a physical Franka Emika arm and showing successful task completion in challenging scenarios (e.g., Figure 5's "dark room" task), the authors demonstrate the practical relevance of their work. The framework's design inherently promotes safety by ensuring that preconditions are met before execution, which reduces the rate of irreversible and thus potentially unsafe actions.


### **Weaknesses**
1. **Latency and Efficiency:** The recursive design, while robust, may introduce latency due to repeated probing, code regeneration, and validation. The paper lacks metrics on planning time or interaction steps, which are key for real-time applications.
2. **Assumptions about “Safe” Probes**: The framework assumes probes preserve task-relevant states, which may not hold in all scenarios (e.g., probing fragile objects). Clarifying the scope of this assumption would strengthen the work.
3. **Originality in Synthesis:** The novelty lies in the effective integration of existing ideas rather than in a fundamentally new algorithm. While NeSyConf and the overall design are impactful, the core components have prior precedent [1, 2, 3].


[1] Liang, J., Huang, W., Xia, F., Xu, P., Hausman, K., Ichter, B., ... & Zeng, A. (2023, May). Code as policies: Language model programs for embodied control. In 2023 IEEE International Conference on Robotics and Automation (ICRA) (pp. 9493-9500). IEEE.

[2] Choi, W., Park, J., Ahn, S., Lee, D., & Woo, H. NeSyC: A Neuro-symbolic Continual Learner For Complex Embodied Tasks In Open Domains. In The Thirteenth International Conference on Learning Representations.

[3] Singh, I., Blukis, V., Mousavian, A., Goyal, A., Xu, D., Tremblay, J., ... & Garg, A. (2023, May). Progprompt: Generating situated robot task plans using large language models. In 2023 IEEE International Conference on Robotics and Automation (ICRA) (pp. 11523-11530). IEEE.

---

> ### Author Rebuttal · Authors · 2025-07-31
>
> Dear Reviewer UanP,
>
> We thank the reviewer for the insightful and encouraging evaluation. We especially appreciate your focus on practical applicability and thoughtful questions about efficiency, safety, and scalability, which we address below.
>
> ---
> ### W1, Q1: Latency and computational cost of recursive validation
> In response to the reviewer's suggestion, we report component-wise computational metrics for the cost-incurring modules of our framework, evaluated under different levels of observability incompleteness in long-horizon RLBench tasks.
> The table below summarizes average planning time, number of probe refinement cycles, and detailed per-task statistics on symbolic tool and LLM usage, including the number of calls and average time per call.
>
> |Observability Level|Planning Time/Task|Probe refinement Cycles/Task|Symbolic tool Calls/Task|Symbolic tool Time/Call|LLM Calls/Task|LLM Time/Call|
> |-|-|-|-|-|-|-|
> |High Incompleteness|517.8s|2.9|16.9|2.3s|26.8|8.3s|
> |Low Incompleteness|264.1s|2.4|11.2|2.1s|19.8|6.0s|
>
> The primary contributor to planning time is LLM response latency, while symbolic tool usage (e.g., SMT and PDDL solvers) adds relatively little overhead.
> While planning incurs a moderate overhead per task, NeSyRo prioritizes reliable task completion over speed. Our focus is on accurately grounding policy code under partial observability rather than minimizing response latency.
> The table below illustrates the trade-off between computational cost and success rate, controlled by the validation threshold $\epsilon$. As $\epsilon$ increases, NeSyRo performs more probe refinement and symbolic checks, resulting in higher planning time but improved task success.
>
> |$\epsilon$|SMT Calls/Task|PDDL Calls/Task|Probe refinement Cycles/Task|Planning Time/Task|SR|
> |-|-|-|-|-|-|
> |0.0|1.3|4.9|0.0|140.9s|55.6%|
> |0.3|3.9|11.6|1.1|167.6s|88.9%|
> |0.6|10.3|18.3|3.5|250.1s|100.0%|
> |0.9|16.0|25.3|5.0|294.7s|100.0%|
>
> NeSyRo adopts a more conservative validation strategy as $\epsilon$ increases, triggering more probe refinement cycles for uncertain steps. This raises planning time (from 140.9s to 294.7s) but significantly improves success rate (from 55.6% to 100%). The results highlight how tuning $\epsilon$ enables a controllable trade-off between reliability and computation, depending on task demands.
>
> ---
> ### W2, Q2: Assumptions and guarantees of probe safety
> We address both the assumptions made about safe probing and the mechanisms that ensure its safety and reversibility below.
>
> - **(1) Scope of the safe probe assumption:**
> We assume that probe policies preserve task-relevant symbolic states by avoiding irreversible changes to the environment during execution. This is supported by our use of deterministic low-level controllers, which allow skills such as pull, pick, or place to execute consistently when preconditions are met. As a result, even fragile objects can be safely manipulated under ideal conditions. However, the framework does not account for stochastic physical effects such as slippage, unstable contact, or object deformation. Handling such dynamics is beyond the current scope of our work and presents a promising direction for future extensions. Further discussion on this aspect can be found in our response to Reviewer xaVS W1, Q4.
>
> - **(2) Safety guarantee mechanism:**
> Our framework does not rely on a manually curated set of “safe” skills. Instead, NeSyRo ensures the safety of each probe policy $\pi_{\text{probe}}$ through neuro-symbolic validation. Specifically, the NeSyConf combines symbolic feasibility $\text{LC}\_{f_n}$ and commonsense plausibility $\text{CSC}\_{f_n}$ for each skill $f_n$. This dual criterion helps detect both logical violations (e.g., unmet preconditions) and contextually unsafe actions. Because only skills with high NeSyConf are executed, and low-confidence probes are recursively refined, the system prevents unsafe code from reaching execution. Thus, the validation process acts as a safeguard, filtering out probe policies likely to cause task failure or irreversible effects.
>
> - **(3) Side effect reasoning:**
> The symbolic planner $\Psi\_{\text{vali}}$ verifies whether a skill $f_n$ satisfies preconditions defined in the PDDL domain, such as “a cup with water must be placed on a low surface.” However, it cannot reason about side effects that are not explicitly modeled, like a cup tipping over when placed too high. To complement this, the Common Sense Confidence $\text{CSC}_{f_n}$, computed by $\Phi\_{\text{vali}}$, leverages retrieved demonstrations and LLM priors to assess contextual plausibility. This allows the system to filter out physically unsafe actions even when such risks are not encoded symbolically.
>
> - **(4) Empirical safety evidence:**
> Our recursive validation strategy effectively minimizes failure modes. In Table 3, we report that NeSyRo executes only 7 irreversible actions under partial observability, compared to 53 with CaP. This empirically confirms that the combined symbolic and commonsense validation significantly improves the safety of information-seeking behavior.
>
> ---
> ### W3: Originality of the proposed framework
>
> As discussed in the Related Work section, we argue that NeSyRo represents a novel neuro-symbolic robot task planning framework that goes beyond a simple combination of existing components. Below, we clarify key distinctions between NeSyRo and the cited methods.
>
> Code as Policies (CaP) and ProgPrompt directly generate control code, but the resulting code often lacks proper grounding in dynamic or partially observable environments due to the absence of explicit feedback integration. This leads to incomplete or non-executable outputs. In contrast, NeSyRo improves code grounding and reliability by incorporating explicit feedback into the generation process.
>
> While NeSyC leverages environmental feedback and a neuro-symbolic framework for embodied task planning, its goal is to construct generalized symbolic knowledge through continual interaction between symbolic tools and LLMs in iterative loops. In contrast, NeSyRo focuses on generating environmentally grounded policy code by recursively gathering missing observational information, thereby strongly regularizing ungrounded execution.
> Specifically, the scoring method in NeSyC, called the HI score, differs from NeSyConf in both purpose and computation. HI measures alignment between transitions and demonstrations, whereas NeSyConf estimates skill feasibility by combining logic-based planning and commonsense priors. The two scores address fundamentally different objectives and are not interchangeable.
>
> NeSyRo's robustness cannot be achieved by simply combining components from prior work. Its key novelty lies in detecting uncertainty during code generation and recursively generating safe probes to recover missing observations before execution (Figure 2, Lines 185–192). This mechanism is essential for reliable performance in dynamic environments, as demonstrated by the empirical results in Section 4.2 and Table 2.
>
> ---
> ### Q3: Scalability and tractability in complex domains
> We address the reviewer's question in two parts:
>
> - **(1) Scalability to complex domains and long-horizon tasks:**
> NeSyRo's modular design allows it to scale from tabletop manipulation to more complex embodied domains, including mobile manipulation in full home environments [1, 2, 3]. In particular, both symbolic tools used for verification and interactive validation can be replaced to suit the target application domain. Recent neuro-symbolic studies [4, 5] have shown that symbolic tools such as SMT solvers and PDDL planners can be adapted to diverse planning problems. NeSyRo is compatible with these approaches and can incorporate domain-specific symbolic tools, such as those for spatial navigation or mobile interaction, without modifying its core architecture [6,7].
> While our experiments focus on tabletop settings, we also include long-horizon tasks in both simulation and the real world, involving multi-step sequences, recursive probing, and dynamic observation recovery, as shown in Table 2 and Table 3 in our paper. These results demonstrate NeSyRo's robustness under uncertainty and its applicability to more complex task structures.
>
> - **(2) Managing complexity in NeSyRo:**
> To address complexity in large environments, NeSyRo adopts two strategies. First, validation is performed at the skill level, using symbolic tools and commonsense estimation based on partial observations. This localized reasoning keeps the process tractable even when the symbolic state space is large. Second, while verification currently applies to the full policy code, it can introduce overhead as task length grows. A natural extension is to decompose tasks into subtasks with local specifications, enabling more efficient verification. This aligns with hierarchical planning approaches [8,9] and is reflected in our framework design.
>
> ---
> We again thank the reviewer. We will incorporate these improvements in the final version.
>
> [1] Puig et al. Virtualhome: Simulating household activities via programs. CVPR 2018.
>
> [2] Kolve et al. AI2-THOR: An Interactive 3D Environment for Visual AI. arXiv 2017.
>
> [3] Shridhar et al. ALFWorld: Aligning Text and Embodied Environments for Interactive Learning. ICLR 2021.
>
> [4] Pan et al. Logic-LM: Empowering Large Language Models with Symbolic Solvers for Faithful Logical Reasoning. Findings of EMNLP 2023.
>
> [5] Lin et al. CLMASP: Coupling Large Language Models with Answer Set Programming for Robotic Task Planning. arXiv 2024.
>
> [6] Lu et al. Neuro-symbolic Procedural Planning with Commonsense Prompting. ICLR 2023.
>
> [7] Zheng et al. JARVIS: A Neuro-Symbolic Commonsense Reasoning Framework for Conversational Embodied Agents. arXiv 2022.
>
> [8] Mirakhor et al. Task Planning for Object Rearrangement in Multi-room Environments. AAAI 2024.
>
> [9] Chen et al. EMOS: Embodiment-aware Heterogeneous Multi-robot Operating System with LLM Agents. ICLR 2025.

---

> ### Comment · Reviewer_UanP · 2025-08-01
>
> Thanks for the authors' responses and clarifications. Will keep my original scores.

---

> > ### Author Response · Authors · 2025-08-02
> >
> > Thank you for your time and for considering our responses. We appreciate your helpful comments and review. We will remain actively engaged during the discussion period, so please feel free to reach out with any further questions or clarifications.

---

### Official Review · Reviewer_xaVS · 2025-06-16

**Clarity:** 4
**Significance:** 3
**Originality:** 4
**Rating:** 5
**Confidence:** 4

**Summary:**

This paper builds on the Code-as-Policies (CaP) paradigm, introducing a framework for neuro-symbolic robotic task planning (NeSyRo), for safer and more reliable plans. As in CaP, an LLM is given a task instruction, and needs to output a policy written in code, that expresses which actions to take in terms of a predefined set of control skills or primitives.
The main features of NeSyRo are the following: first, it introduces two processes to refine the generated code, verification and validation. The verification uses an SMT solver to check that the policy code satisfies some LLM-generated specifications and iteratively re-generates the policy taking into account feedback from the solver until this step is successful. The validation checks skill-by-skill if there exists a symbolic plan from the current state to the goal which contains that skill (Logic Confidence), and also if the LLM assigns a cumulative log probability (normalised somehow) over the code for the skill higher than a certain threshold (given some proper context). Second, in case of failed validation, it uses a safe probe code policy to gather information that might help resolving the validation step, which can fail due to insufficient contextual information. Finally, the framework uses a recursive resolution strategy, where the main pipeline to generate the code policy undergoes verification and validation, and triggers the generation of a safe probe in case of failed validation; the probe undergoes a similar verification and validation and can recursively trigger other safe probes, forming a policy tree. Experiments on simulated RLBench and real-world settings convincingly prove the strength of the method when compared to a range of strong CaP-like baselines.

**Questions:**

1. Can you provide a formal definition of what you intend with “dynamic environment”?
2. Is the safe probe ever taking irreversible actions? And does it share the same action space (set of code primitives) of the main pipeline?
3. In the validation step, it seems that you need to perform some sort of lookahead validation, as the latest observation (and the full history) that you have at disposal is used to assess the confidence that all the skills in the plan will be correct. Is this the case? If so, how does the common sense confidence take into account the assessment of whether an action will be plausible n steps in the future?
4. Can the framework as-is handle action failures due to imperfect low-level execution (i.e. due to stochasticity in the environment)? My understanding is that the safe probe can gather information in a POMDP, but that doesn't necessarily translate to robustness to stochasticity. Could some sort of replanning mechanism (e.g. Model-Predictive Control), either at the NeSyRo level, or at the level of the synthesised policy help?
5. Can you provide a more detailed description of the verification step, with a simple example of what a correct task specification might look like, which SMT solver you are using and how the feedback might look like? Do you assume to have a PDDL domain with operators matching the code skills/primitives, in order to do this step? And how do you ground PDDL in the environment? Do you have a ground-truth representation function of every (or at least the initial) state? Overall I think the assumptions of this step should be clarified in the main text and the answer to this question could be added as a section of the appendix.
6. If PDDL is used so extensively, why not plan directly solely via PDDL? Is the main difference that in PDDL one cannot express the parameter for the skills, and as such it could decide which sequence of actions to take, but not which parameters these actions should have?
7. Can you explain in more detail how the retrieved demonstrations are collected? As I understand it, your framework doesn’t include a training phase, so is it fair to assume having at disposal demonstrations to retrieve at test time?
8. How much overlap is there between the work cited for CaP w/ Lemur baseline and your implementation of the Neuro-symbolic Code Verification? Similar question for CaP w/ CodeSift and your verification and especially validation implementations.

**Ethical Concerns:**

["NO or VERY MINOR ethics concerns only"]

**Final Justification:**

The discussion with the authors clarified many details of the proposed method. The authors made clear plans in their final acknowledgement to include the additional explanations in the paper. Overall it's a very strong and solid paper, which will probably be strengthened by including these discussions in the camera-ready version.

**Limitations:**

Limitation section should be definitely expanded (currently two lines) to discuss all the assumptions they make (see weaknesses and questions).

**Paper Formatting Concerns:**

No concerns.

**Quality:**

4

**Strengths And Weaknesses:**

Strengths
- Complete and sensible system to increase the reliability of Code-as-Policies approaches; the method is well designed, with original solutions to all the sub-problems the authors had to tackle to make it work
- Paper is clearly written and manages to present a complex method in an understandable way
- Good experimental setup, very nice baselines, convincing main results
- Helpful ablations on the usefulness of both verification, validation and impact of LLM model on the code generation and the LLM-based validation confidence score

Weaknesses
- Not all assumptions are clear, it would be good for the paper to explain better exactly all the extra machinery that one needs to make this approach work with respect to what vanilla CaP requires
- The limitations section is virtually non-existent, while it should reflect for example when are the assumptions of the point above reasonable to assume.
- Some details and citations are missing, like which SMT solver the authors use (a footnote would do).
- The problem formulation should include some notes about how the action space looks like in the CaP case, how is the state representation handled (how high or low level it is) and what (if any) is the difference between actions and skills.
- It’s not really clear how much the baselines considered overlap with the method proposed.

Other notes
- In the camera-ready version, please make sure to cite the appendix sections in the main text. I understand it has been submitted later as supplementary material, so it’s okay at this stage not to have that.

---

> ### Author Rebuttal · Authors · 2025-07-30
>
> Dear Reviewer xaVS,
>
> We appreciate your clear articulation of both the strengths and the areas for improvement, which we address below.
>
> ---
> ### W1, W2: Missing assumptions, additional components, and limitations discussion
> Below is a concise response structured around three aspects of the reviewer's concerns:
>
> - (1) Assumptions & resources
>     - Partial observability: Initial symbolic observation is given; additional predicates are recovered online via perception (e.g., VLM).
>     - Deterministic low-level control.
>     - Task coverage: the goal can be expressed as a sequence of skills in $\mathcal{D}$.
>     - Resources: domain knowledge $\mathcal{D}$, demo set $\mathcal{E}\_{\text{demo}}$ (RAG), observation buffer $o\_{\le t}$, SMT solver (Z3), PDDL planner (Fast Downward), validation LLM for commonsense scoring.
>
> - (2) Extra machinery beyond vanilla CaP
>     - Neuro-symbolic verification & validation
>     - Safe probe synthesis when $\text{NeSyConf}<\epsilon$, with probes undergoing the same recursive verification & validation process.
>     - Skill refinement: only the offending skill suffix is regenerated.
>
> - (3) Limitation
>     - NeSyRo assumes deterministic execution and does not yet handle stochastic failures, as noted by Reviewer xaVS Q4. Execution-time replanning (e.g., MPC) is a promising future direction.
>
> ---
> ### W3, Q5: Underspecified verification and PDDL assumption details
> We clarify the verification step and assumptions below.
>
> - (1) Verification process: We use the Z3 SMT solver to verify policy code $\pi_{\text{main}}$ satisfies a task specification $\mathcal{T}_{\text{spec}}$, including a high-level plan and parameter constraints. If verification fails, feedback is generated through the LLM and SMT solver, as detailed in our response to Reviewer nsGi Q4. The framework assumes that specification skills align with the PDDL domain knowledge, but it remains robust even when generation errors cause mismatches, as discussed in our response to Reviewer nsGi Q1. An example of the task specification is shown below.
> ```json
> {
>   "plan": [
>     {"skill": "pull", "args": {"target": "middle_drawer", "dist": , ...}},
>     {"skill": "pick", "args": {"target": "dice"}, ...},
>     {"skill": "place", "args": {"target": "top_drawer"}, ...},
>     ...
>   ],
>   "constraints": {
>     "0": {"dist": [0.05, 0.15]},
>     ...
>   }
> }
> ```
> - (2) PDDL grounding assumptions: We assume that the initial symbolic observation is externally provided and partially aligned with the ground-truth state, containing a subset of task-relevant predicates. This controlled incompleteness defines partially observable scenarios in a reproducible way. During execution, missing or ambiguous predicates are recovered via safe probes, with visual inputs processed by a VLM to extract symbolic information. Thus, only partial ground-truth access is assumed initially, with all further updates handled by perceptual modules.
>
> We will include these details and the SMT solver name in the final version.
>
> ---
> ### W4: Unclear action/skill/state
> An action is an atomic low-level primitive (e.g., move\_to(x, y, z), grasp( )) that controls the robot in continuous space. By contrast, a skill is a higher-level symbolic unit that parameterizes and composes a sequence of such actions to achieve a subgoal (e.g., pull(drawer, dist=0.1) may invoke multiple move_to and grasp calls). Under this view, the action space is implicitly defined as a continuous control space over which the robot operates, while the skill space is defined symbolically by the set of parameterized function calls. The policy code $\pi_{\text{main}}$ consists of a sequence of skills $(f_0, f_1, \ldots, f_N)$.
>
> The state is symbolic and structured. Observations are stored as symbolic facts (e.g., is_locked(“top_drawer”)) in observation buffer $o_{\leq t}$. These symbolic observations are incrementally constructed through interaction (e.g., via safe probe).
>
> ---
> ### W5, Q8: Ambiguous overlap with CaP baselines
>
> While the baselines are adapted from CaP, their mechanisms differ substantially from ours.
>
> - (1) CaP w/ Lemur:
> Both CaP w/ Lemur and NeSyRo perform verification by generating full policy code and checking it with an SMT solver.
> However, CaP w/ Lemur repeatedly regenerates the entire code to match a fixed specification, whereas NeSyRo uses chain-of-thought prompting to induce the specification and performs skill-level patching when verification fails. This enables a more modular and efficient refinement process.
>
>
> - (2) CaP w/ CodeSift:
> CaP w/ CodeSift and NeSyRo both aim to improve the reliability of LLM-generated code through verification and validation, but differ fundamentally in their approach. CodeSift relies on static, multi-stage LLM-based checks without symbolic reasoning. In contrast, NeSyRo employs a neuro-symbolic strategy, combining CSC and LC, for interactive validation and environment-aware refinement under partial observability.
>
> ---
> ### Q1: Define “dynamic environment” formally
> We call the environment dynamic when executing at least one action can change the symbolic state. Let $\mathcal{S}$ be the set of states, $\mathcal{A}$ the set of atomic motion primitives, and $T$ the transition function. The environment is dynamic if and only if
>
> $\exists\ s \in \mathcal{S}, a \in \mathcal{A} \quad \text{s.t.} \quad T(s,a) \neq s$
>
> That is, some atomic action modifies the symbolic state. Let $f$ be the domain-level library of parameterized skills, each implemented as a finite sequence of actions from $\mathcal{A}$, and thus inherits this property.
>
> ---
> ### Q2: Probe safety and action space consistency
> The safe probe policy is unlikely to take irreversible actions and shares the same action and skill space as the main policy.
>
> - (1) Irreversible actions: Like the main policy, the safe probe is verified and validated to prevent unsafe actions. As shown in Table 3, NeSyRo-Complete setting (fully observable environment, no probe needed) results in 6 irreversible actions (IA), while NeSyRo with probe results in only 7. This minimal increase shows that safe probe rarely causes irreversible effects, unlike CaP, which leads to 53 IA.
>
> - (2) Shared action space: The safe probe is generated and checked using the same pipeline as the main policy. It uses the same set of parameterized skills from the domain knowledge $\mathcal{D}$ and follows identical code generation and refinement procedures.
>
> ---
> ### Q3: CSC's future plausibility estimation
> Our framework performs sequential n-step lookahead validation using only the current observation history $o_{\le t}$, assuming prior skills are validated. The process begins with the first skill $f_0$ with confidence $\text{NeSyConf}_{f_0}$ from CSC and LC. Once $f_0$ is validated, we symbolically simulate its effects to generate a predicted observation $\hat{o}_1$ to validate $f_1$. This recursive process continues for subsequent skills.
>
> Importantly, CSC evaluates the plausibility of each skill using $o_{\le t}$ and retrieved demos. It does not perform explicit multi-step foresight. Instead, lookahead is realized by the framework as a whole: each step's predicted observation feeds into the validation of the next skill. This symbolic unrolling enables multi-step assessment.
>
> ---
> ### Q4: Handling stochastic failures
> NeSyRo is designed to handle partial observability by leveraging neuro-symbolic validation and recursive safe probing. This process recovers missing observations, improving policy grounding and task reliability. However, NeSyRo does not address low-level stochastic failures (e.g., mis-grasps). These failures are outside the scope of our framework, which assumes deterministic execution once the policy is validated.
>
> We agree that incorporating replanning mechanisms, such as Model Predictive Control (MPC), could help recover from execution-time failures without restarting the entire plan. Future work could extend NeSyRo beyond symbolic grounding by exploring such mechanisms.
>
> ---
> ### Q6: Limitations of PDDL-only planning
> PDDL alone cannot express continuous parameters and requires fully observable states, making effective planning under partial observability challenging. Specifically, PDDL has two critical limitations:
>
> - (1) Limited parameter expressiveness: PDDL cannot represent continuous parameters. LLM-generated code addresses this limitation by accurately adjusting these continuous parameter values.
>
> - (2) Inability to plan under partial observability: PDDL assumes full observability, so it cannot handle uncertainty (e.g., unknown drawer states). Our framework uses safe probes to check uncertain conditions (e.g., is a drawer locked). This active observation recovery allows continual refinement of the plan based on updated environmental information.
>
> Experiments show that combining LLMs with PDDL overcomes these limits and improves task success under partial observability.
>
> ---
> ### Q7: Assumptions behind demo retrieval
> Our framework does not involve any training phase. Instead, it retrieves demonstrations at test time from a pre-constructed, synthetic library using retrieval-augmented generation, conditioned on the current observation and instruction. As described in Appendix C.2, all demonstrations are generated by GPT-4o without environment execution or human supervision. We synthesized 500 demonstrations covering 15 skill types. The collection proceeds as follows:
> - (1) Domain definition: A domain-level PDDL is defined to capture symbolic transitions (e.g., drawers, bins).
>
> - (2) Context specification: We instantiate concrete symbolic states (e.g., “the third drawer is locked, a cup is on the table”).
>
> - (3) Instruction \& plan generation: GPT-4o generates a natural language instruction and the corresponding symbolic plan.
>
> - (4) Demonstration structuring: Plans are segmented into skill-level demonstrations with pre/post observations and success labels.
>
> ---
> We again thank the reviewer. We will incorporate these improvements in the final version.

---

> > ### Comment · Reviewer_xaVS · 2025-08-03
> > **Rebuttal acknowledgement**
> >
> > I thank the authors for their comprehensive answer. Most of the points I raised were well addressed and clarified. I have some follow-up questions and comments from our discussion, which I report below.
> >
> > **1. Observations are symbolic**
> >
> > I realised that I misunderstood the nature of the observations $o$ in your framework and assumed that were image-based; now the validation step is much clearer (e.g., Q3 and Q7). I suggest you to clarify this in section 3.1. Also, does this mean that the timestep $t$ that you report is measured in terms of skills executed (i.e., each skill lasts 1 timestep by definition) and not some lower level timestep?
> >
> > ** 2. Code skills vs PDDL actions**
> >
> > Your answer to Q6 is really helpful. Is this way of thinking about the two correct?
> > - **Code skills:** how to implement a skill in terms of low-level primitives (move_to, grasp, etc.)
> > - **PDDL actions:** logical preconditions and effects that some skill has on the symbolic state of the environment
> >
> > And so if I get it correctly, you're saying that the code skills enable to actually implement the code policy and thus are much more expressive? If that was the point you were making, that makes sense to me.
> >
> > **3. Computing Logic Confidence (LC) score in partial observations**
> >
> > You mention that PDDL cannot handle uncertainty, yet, if I am not mistaken, you use PDDL planners to compute the the LC score. What happens when the symbolic state is not fully determined? Do you directly give a score of 0, or do you attempt to synthesize a plan with a PDDL planner in some way?
> >
> > **4. Synthetic demonstrations**
> >
> > Are the demonstrations unrolled for multiple steps (like sort of PDDL trajectories), or are they demonstrating a single skill transition each? My understanding is the latter. In such case, maybe demonstrations is a confusing name, as in RL/Imitation Learning usually demonstrates a full trajectory from a behavioral policy.
> >
> > **5. Limitations**
> >
> > In W1/W2 I was trying to make the point that everything that you assume available on top of CaP can lead to potential limitations of the method. For example, as the PDDL domain is assumed given in NeSyRo, my understanding (please correct me if I'm wrong) is that you can only define skills that have a correspondent PDDL action, whereas CaP-methods non reliant on PDDL for validation can potentially generate skills beyond the basic ones defined in the domain. I think it's a point worth mentioning in the limitations.
> >
> > Thank you again for engaging in the discussion and the good work.

---

> > > ### Author Response · Authors · 2025-08-03
> > >
> > > Thank you for the thoughtful follow-up and for taking the time to re-engage with our work. We're glad to hear that many of your initial points were addressed and clarified. We greatly appreciate the constructive nature of your questions, and we respond to each point below.
> > >
> > > ---
> > > ### 1. Observations are symbolic
> > > We appreciate the reviewer’s clarification. We will revise Section 3.1 to explicitly state that observations in our framework are symbolic, composed of structured predicate-based representations. As the reviewer mentioned, timesteps in our experiments are measured at the skill level, where each execution of a single skill function corresponds to one timestep.
> > >
> > > ---
> > > ### 2. Code skills vs PDDL actions
> > > The reviewer's interpretation of the distinction between PDDL actions and code skills is correct. It is also accurate that code skills are more expressive, as they enable the actual construction of executable policy code. This expressiveness comes from their ability to encode conditional logic, and direct interaction with the robot’s control API, allowing for richer and more flexible behavior than what can be represented through symbolic abstractions alone.
> > >
> > > ---
> > > ### 3. Computing Logic Confidence (LC) score in partial observations
> > > We use a PDDL planner to compute the LC score even in partially observable states. Since we only need to assess the feasibility of an individual skill (rather than generate a complete task plan) the planner can be used in a task-agnostic way. Each candidate skill is defined in the domain knowledge with explicit preconditions and effects. By treating the effect of the skill as a temporary goal state, we allow the planner to check whether the current symbolic state satisfies the required preconditions. This provides a symbolic estimate of the skill's feasibility. As a result, we obtain a binary LC score (0 or 1) for each skill, since the PDDL planner determines feasibility based on whether the skill's preconditions can be satisfied.
> > >
> > > ---
> > > ### 4. Synthetic demonstrations
> > > Thank you for the helpful clarification. The reviewer is correct in observing that each demonstration in our framework corresponds to a single skill execution, rather than a full trajectory. To avoid potential confusion, we will refer to them as "single-skill demonstrations" in the final version of the paper. This terminology aligns with prior work where demonstrations are defined not by their temporal length but by their role as reusable units for learning or inference [1, 2]. Since our demonstrations are used to validate individual skills across diverse contexts, we follow the same convention.
> > >
> > > ---
> > > ### 5. Limitations
> > > We sincerely appreciate the reviewer's insightful observation regarding the limitations of our framework. This feedback has helped us better articulate the current scope and future direction of NeSyRo.
> > >
> > > It is indeed correct that the current version of NeSyRo relies on a predefined PDDL domain, which limits validation to skills with symbolic representations. This limitation is closely tied to our use of binary PDDL, where LC is either 0 or 1 based on strict precondition satisfaction.
> > >
> > > However, the NeSyConf formulation is designed to allow LC and CSC to complement each other, similar to how SayCan [3] combines LLM scores with affordance function. As we mentioned in Conclusion Section, for future direction, enabling a non-binary LC (e.g., probabilistic PDDL [4]) could extend validation to skills not explicitly defined in the domain knowledge.
> > >
> > > ---
> > > Thank you again for the continued engagement and the thoughtful discussion throughout the rebuttal phase. We remain committed to providing thorough responses during the discussion period. Please feel free to raise any additional questions or points of clarification.
> > >
> > > [1] Rana et al. “Residual Skill Policies: Learning an Adaptable Skill‑based Action Space for Reinforcement Learning for Robotics”. CoRL 2022.
> > >
> > > [2] Auddy et al. “Continual Learning from Demonstration of Robotics Skills”. Robotics and Autonomous Systems, 2023.
> > >
> > > [3] Ahn et al. "Do as i can, not as i say: Grounding language in robotic affordances". CoRL 2022.
> > >
> > > [4] Younes et al. PPDDL 1.0: An extension to PDDL for expressing planning domains with probabilistic effects. 2004.

---

> > > > ### Comment · Reviewer_xaVS · 2025-08-04
> > > >
> > > > I thank the authors for their follow-up answer, which clarified all the remaining points. So far, I find my initial score to well reflect my judgement of the paper, but I will remain with an open mind going into the discussion with other reviewers and AC, and will consider this discussion in the final rating of the paper.

---

> > > > > ### Author Response · Authors · 2025-08-04
> > > > >
> > > > > We sincerely thank the reviewer for the thoughtful engagement and for considering our clarifications throughout the rebuttal. We truly appreciate the opportunity to participate in this discussion.

---

### Official Review · Reviewer_nsGi · 2025-06-17

**Clarity:** 2
**Significance:** 2
**Originality:** 2
**Rating:** 5
**Confidence:** 3

**Summary:**

This paper explores the use of LLMs to generate directly executable code for task planning and control in embodied agents. In the proposed framework, the generated code undergoes holistic verification through symbolic verification, followed by sequential validation of individual functions via interactive validation. The method is empirically shown to be effective in dynamic and partially observable environments.

**Questions:**

1. In section 3.3, is it possible that the task specification is unclear or incorrect for the code generation?

2. In section 3.3, how should we interpret "a sequence of skills with their parameters and required library"?

3. In section 3.3, how should we understand "domain knowledge $D$" consisting of available skills, object types, and environment-specific constraints?


4. In symbolic-based code verification, how should we interpret the feedback $\mathcal{F}\_{\mathrm{veri}}$ from the verification tool, and what does it look like?


5. In the neuro-symbolic confidence score, how should we understand "past demonstration," and why is it important?


6. How should we interpret $\mathcal{F}\_{\mathrm{csc}}$ and $\mathcal{F}\_{\mathrm{ls}}$  in Equation 6?

**Ethical Concerns:**

["NO or VERY MINOR ethics concerns only"]

**Final Justification:**

During the rebuttal period, the authors have addressed my concerns regarding computational cost and the importance of past demonstrations. They provided detailed explanations and responded thoroughly to my questions.

**Limitations:**

This method may excessively rely on external tools and explorations in the environment, which could raise concerns about efficiency.

**Paper Formatting Concerns:**

N.A.

**Quality:**

2

**Strengths And Weaknesses:**

## strengths


1. Unlike prior works that directly generate executable code without rigorous verification, this method introduces a two-stage verification process: holistic symbolic verification and component-wise interactive validation. This decomposition is designed to address the challenges posed by dynamic and partially observable environments.

2. During interactive validation, each component function is first assessed using a newly defined neuro-symbolic confidence score, which determines whether refinement through direct interaction with the environment is necessary. This interaction enables exploration and adaptation, enhancing the quality of the resulting executable code used as policy. This mechanism marks a key distinction from previous approaches.



## weaknesses


1. At each step, the symbolic verification tool is invoked to identify potential violations in the generated code. However, it is unclear whether the use of such external tools incurs significant computational cost. In the context of using LLMs for task planning, it is generally preferable to rely on the model itself for error correction, rather than heavily depending on external verification tools. I strongly recommend that the authors introduce a metric to quantify the cost associated with invoking these tools during the verification process.

2. The neuro-symbolic confidence score comprises two components: common sense confidence and logic confidence. The input to the common sense confidence includes past demonstrations, but the importance of this design choice is not clearly justified. I suggest that the authors provide empirical evidence supporting the contribution of this component and offer an intuitive explanation for its inclusion.

3. To enhance performance in dynamic and partially observable environments, the framework introduces a probe policy that explores the environment based on the current policy. However, it is unclear whether generating this probe policy is computationally efficient and how its safety is ensured during interaction. Given that exploration via the probe policy plays a critical role in improving the final policy, I am concerned about the potential safety risks and recommend a more detailed discussion on how these are mitigated.

4. During interactive validation, excessive interaction between the probe policy and the environment is undesirable. Therefore, it is important to define a metric that quantifies the cost of refining the policy through such interactions.

5. A minor issue is that some descriptions in the paper are overly abstract or ambiguous, particularly regarding certain notations. I recommend that the authors include concrete examples in the main text to clarify these concepts. Some of these ambiguities have been noted in the questions section.

---

> ### Author Rebuttal · Authors · 2025-07-29
>
> Dear Reviewer nsGi,
>
> We are grateful for the reviewer's detailed assessment and for noting the strengths of our approach. Below, we address each of the specific concerns and questions in detail.
>
> ---
> ### W1, W4: Metric for external tool cost and probe cost, LLM-only correction
> In response to the reviewer's suggestion, we provide an analysis of the computational cost associated with symbolic tool usage and probe execution during refinement. NeSyRo employs two symbolic tools:
>
> - (1) an SMT verifier that checks policy code compliance with the task specification.
> - (2) a PDDL planner that assesses skill feasibility during validation.
>
> These tools are used for code verification and skill grounding. When the NeSyConf score for a skill falls below a threshold $\epsilon$, NeSyRo triggers an additional safe probe refinement cycle involving further symbolic verification and validation. As a hyperparameter, $\epsilon$ controls the frequency of symbolic tool usage, balancing computational cost and reliability. To quantify this trade-off, we varied $\epsilon$ and measured the average number of SMT solver calls, PDDL planner calls, and safe probe executions per task.
>
> |$\epsilon$|SMT Calls/Task|PDDL Calls/Task|Probe Exec./Task(max. 5)|SR|
> |-|-|-|-|-|
> |0.0|1.3|4.9|0.0|55.6%|
> |0.3|3.9|11.6|1.1|88.9%|
> |0.6|10.3|18.3|3.5|100.0%|
> |0.9|16.0|25.3|5.0|100.0%|
>
> As $\epsilon$ increases, symbolic tool usage rises, improving grounding and task success. Lower values reduce cost but may compromise performance. Thus, $\epsilon$ serves as a tunable parameter that balances tool cost and policy reliability based on task requirements. Correspondingly, the number of probe executions also increases with $\epsilon$, as more skills fall below the threshold and require refinement.
>
> Prior works [1, 2] show that LLMs often miss semantic errors due to hallucination. NeSyRo mitigates this through explicit symbolic reasoning, preventing irreversible actions and improving success rates, as shown in Table 3. While optimizing computational cost is an important direction for future work, this paper focuses on ensuring reliability and robustness during task execution. We hope this design choice and trade-off analysis address the reviewer's concerns, and we will include the analysis and tool usage details in the final version.
>
> ---
> ### W2, Q5: Role and impact of past demonstrations
> As described in Section 3.4 (lines 173–175), past demonstrations are used in the CSC component to reduce hallucinations and ground the LLM's commonsense reasoning in the environment. While LLMs excel at general reasoning, their difficulty in producing grounded outputs under partial observability is a well-known limitation in LLM-based embodied control [3, 4].
>
> To address this, NeSyRo retrieves demonstrations $\mathcal{E}\_{\text{demo}}$ with similar context to skill $f_n$ and inputs them to the validation LLM $\Phi\_{\text{vali}}$ for CSC computation (Eq. 4).
> These demonstrations act as environment-specific priors to assess the plausibility of $f_n$ under $o_{\leq t}$ and $g$, helping reduce hallucination and improve grounding. Recent LLM-based frameworks have similarly used retrieval to enhance contextual grounding in planning tasks [3, 4].
>
> The table below presents an ablation of $\mathcal{E}\_{\text{demo}}$ on long-horizon RLBench tasks under partial observability. We compare NeSyRo with a variant that disables demonstration retrieval in the CSC module. Removing $\mathcal{E}\_{\text{demo}}$ results in notable drops in both SR and GC, highlighting the importance of contextual demonstrations in reliably estimating CSC.
>
> |Method|SR|GC|
> |-|-|-|
> |NeSyRo|45.0%|54.3%|
> |NeSyRo w/o $\mathcal{E}\_{\text{demo}}$|40.0%|44.6%|
>
> ---
> ### W3: The efficiency and safety of probe policy
> We address the reviewer's concerns in two parts: (1) the computational cost during probes, and (2) probe safety under partial observability.
>
> - **(1) Computational cost:**
> Unlike LLM-Planner, which regenerates full policies upon failure, NeSyRo refines only low-confidence skills through targeted safe probes. Symbolic tools are invoked selectively, only when NeSyConf falls below the threshold $\epsilon$, with minimal overhead. This localized refinement leads to significantly fewer interactions with the environment. For example, in long-horizon tasks, NeSyRo completes execution in 18.2 steps on average, compared to 26.8 for LLM-Planner, confirming both computational and interaction efficiency.
>
> - **(2) Safety of probe execution:**
> Each probe policy $\pi_{\text{probe}}$ undergoes the same verification and validation process as the main policy $\pi_{\text{main}}$.
> If a probe lacks sufficient grounding, NeSyRo recursively generates sub-probes, each of which is also verified and validated prior to execution. This recursive structure ensures that all code executed in the environment is symbolically consistent and grounded. As shown in Table 3, NeSyRo significantly reduces irreversible actions during execution, demonstrating its robustness under partial observability.
>
> ---
> ### W5, Q3, Q4, Q6: Clearer definitions and concrete examples
> To address concerns about abstraction, we provide concrete explanations for three core components.
>
> 1. **Domain knowledge $\mathcal{D}$.** In our framework, $\mathcal{D}$ denotes domain knowledge used for verification and validation. It consists of:
>
> - Available skills: Executable operations (e.g., pick, push, pull), which are implemented as parameterized function calls in the policy code.
> - Object types: Symbolic object classes (e.g., mug, drawer, handle).
> - Environment-specific constraints: Symbolic predicates (e.g., `is_open(drawer)`) representing action preconditions and effects.
> These constraints are encoded in the PDDL domain file and used during symbolic verification and validation.
>
> 2. **Verification feedback $\mathcal{F}\_{\text{Veri}}$.** The symbolic verifier checks whether the generated policy code satisfies the task specification, which includes a high-level plan and partially instantiated parameters. If verification fails, the SMT solver returns constraint violations, which the LLM interprets into fine-grained feedback $\mathcal{F}\_{\text{Veri}}$. This feedback, expressed in natural language (e.g., parameter mismatch: `drawer = "middle_drawer" is incorrect; expected: "top_drawer"`), enables the LLM to revise only the erroneous parts.
>
> 3. **Validation feedbacks $\mathcal{F}\_{\text{CSC}}$ and $\mathcal{F}\_{\text{LC}}$.** During validation, if the NeSyConf score falls below a threshold $\epsilon$, feedback from both components is used to generate a safe probe policy $\pi_{\text{probe}}$.
>
> - $\mathcal{F}\_{\text{CSC}}$: Generated by prompting the LLM with the current observation and instruction, providing natural language explanations for implausible skills (e.g., `"There is no need to open the drawer."`)
>
> - $\mathcal{F}\_{\text{LC}}$: Produced by the symbolic planner when a skill's preconditions are not met (e.g., `"(drawer1-open) not satisfied."`)
>
>   These feedbacks guide $\pi_{\text{probe}}$ to recover missing observations or satisfy unmet preconditions.
>   This process is repeated recursively until all skills are grounded and validated as described in Section 3.4 and Figure 2.
>
> ---
> ### Q1: Unclear or incorrect task specification for code generation
> Under partial observability, task specification $\mathcal{T}\_\text{spec}$ may be unclear or incorrect due to incomplete observations. NeSyRo is explicitly designed to handle such cases via recursive validation and refinement. Instead of assuming $\mathcal{T}\_\text{spec}$ is complete, NeSyRo validates each skill in $\pi\_\text{main}$ via Neuro-Symbolic Code Validation. If a skill’s NeSyConf < $\epsilon$, NeSyRo invokes a safe probe, updates the observation buffer, and refines the policy and task specification.
>
> This process is well illustrated in Figure 5 of our paper. Initially, $\pi\_\text{main}$ contains an incorrect high-level plan due to missing observations (e.g., unknown drawer lock states). Through recursive validation, NeSyRo detects these issues and executes targeted safe probes (e.g., checking the lock state of each drawer). As new observations are collected, NeSyRo refines the parameters or structure of the plan, leading to a grounded and executable policy. This example shows how NeSyRo incrementally corrects $\mathcal{T}_{\text{spec}}$ via recursive refinement.
>
> ---
> ### Q2: Meaning of skill sequences, parameters, and required library
> This phrase refers to the structure of the initial Python policy code $\pi_{\text{main}}$. The code consists of sequential calls to predefined skill functions (e.g., pick, place, pull), each parameterized with task-relevant arguments such as object identifiers, axes, or distances. These parameters are derived from $\mathcal{T}_{\text{spec}}$, which encodes subgoals and symbolic constraints.
>
> In some cases, additional computation is needed to determine skill parameters. To support this, the LLM includes import statements for external Python libraries when necessary. For example, libraries such as numpy may be used for distance computation or coordinate transformation. This is one representative case, but the choice of library depends on the specific context and computation required for executing each skill. Thus, "required library" broadly refers to any module needed for correct code execution.
>
> ---
> We again thank the reviewer. We will incorporate these improvements in the final version.
>
> [1] Ji et al. Survey of Hallucination in Natural Language Generation. ACM Computing Surveys 2023.
>
> [2] Tyen et al. LLMs cannot find reasoning errors, but can correct them given the error location. ACL 2024.
>
> [3] Mon-Williams et al. Embodied large language models enable robots to complete complex tasks in unpredictable environments. Nature Machine Intelligence 2025.
>
> [4] Wang et al. Voyager: An Open-Ended Embodied Agent with Large Language Models. arXiv 2023.

---

> > ### Comment · Reviewer_nsGi · 2025-08-04
> >
> > Thank you for your detailed responses and clarifications. I am glad to see the newly provided experimental results regarding computational cost and the importance of past demonstrations. I am willing to increase my score to the accept level.

---

> > > ### Author Response · Authors · 2025-08-04
> > >
> > > We sincerely thank the reviewer for their thoughtful reassessment and for considering our additional results regarding computational cost and demonstration usage. We greatly appreciate the willingness to increase the score and the constructive engagement throughout the review process.

---

### Official Review · Reviewer_rKZ1 · 2025-06-30

**Clarity:** 4
**Significance:** 3
**Originality:** 2
**Rating:** 5
**Confidence:** 3

**Summary:**

The authors propose an LLM code planner for robotic tasks that utilizes recursive functions and symbolic code verification. After verification the method uses a validation step to predict task success using common sense and confidence metrics. Lower confidence code triggers a probing phase to gather more observations prior to adjusting the policy. This continues recursively.

**Questions:**

Are there results for ablating the recursive element of the method?

What are the failure modes of your method on these tasks?

**Ethical Concerns:**

["NO or VERY MINOR ethics concerns only"]

**Limitations:**

yes

**Quality:**

4

**Strengths And Weaknesses:**

Strengths:
- Clearly communicated contribution, method, and results
- Thorough experimental results on multiple tasks and against multiple baselines and with multiple ablation studies
- Precise and high quality figures

Weaknesses:
- Limited to binary confidence
- Domain specific aspects of method limit generalizability
- Leans heavily on empirical results without theoretical justification for method
- I'm unsure whether the baselines the authors choose are comprehensive

---

> ### Author Rebuttal · Authors · 2025-07-28
>
> Dear Reviewer rKZ1,
>
> We thank the reviewer for their detailed feedback and for recognizing the strengths of our work. Below, we provide point-by-point responses to the raised weaknesses and questions.
>
> ---
> ### W1: Limited to binary confidence
> As discussed in the Conclusion as a limitation, the current version of NeSyRo uses a binary Logic Confidence (LC), which indicates whether a symbolic planner (e.g., a PDDL planner) confirms the feasibility of a skill under the current observation. While this binary LC is sufficient for triggering safe probes in our neuro-symbolic validation pipeline, we leave the exploration of continuous formulations of LC as future work.
>
> By incorporating probabilistic reasoning, such as Probabilistic PDDL [1], NeSyRo could represent graded estimates of symbolic feasibility, rather than a hard 0/1 outcome. This would enable more adaptive behavior in probe scheduling and policy refinement.
>
> For example, in open-ended long-horizon tasks involving multiple sequential or compositional subgoals, a continuous LC could help prioritize which skill to validate first based on confidence levels, such as probing a low-confidence skill before executing one with moderate confidence. This allows the agent to recover critical preconditions earlier, reducing redundant probes and improving overall efficiency.
>
> We would like to clarify that the current use of binary LC in NeSyRo does not compromise the framework's effectiveness. Since the symbolic validation module relies on a PDDL planner that yields discrete feasibility outcomes, binary LC is sufficient to determine whether a skill $f_n$ requires further observation and reliably triggers safe probe generation during the Neuro-Symbolic Code Validation phase.
>
> ---
> ### W2: Domain specific design limits generalizability
> While our framework is primarily evaluated in the context of robotic task planning, its architecture is not domain-specific. NeSyRo can generalize to a broad range of planning problems, including non-robotic tasks such as logical puzzle solving or workflow synthesis. Adapting to these domains may require changes to the symbolic interface, but the core verification and validation paradigm remains applicable.
>
> Although we use a PDDL-based planner in our current implementation, the symbolic verification module is modular and can support alternative tools such as ASP-based planners. This flexibility allows our framework to interface with symbolic formalisms appropriate for different environments and planning abstractions.
>
> Our method requires domain knowledge $\mathcal{D}$ describing available skills, object types, and symbolic predicates. However, this is a general requirement in symbolic reasoning systems, not a limitation specific to our approach. NeSyRo mitigates this dependency by incorporating commonsense reasoning through LLMs. The Common Sense Confidence (CSC) component complements symbolic logic with contextual priors and helps infer or refine $\mathcal{D}$, reducing the need for fully specified domain definitions.
>
> Our experiments cover both RLBench and real-world scenarios across diverse task types and configurations, as shown in Tables 2 and 3, demonstrating that the same architecture performs robustly without any domain-specific tuning.
>
> ---
> ### W3: Heavily empirical with limited theoretical justification
> We emphasize that while NeSyRo is supported by extensive experiments demonstrating its effectiveness, it is not purely empirical. Its core mechanisms are grounded in formal symbolic reasoning and verification principles. Specifically, the two central components of NeSyRo, Neuro-Symbolic Code Verification and Neuro-Symbolic Code Validation, are implemented using rule-based symbolic tools and offer the following theoretical justifications:
>
> - **(1) In the code verification phase,** we use an SMT solver [2] to formally check whether the generated policy code satisfies the high-level task specification. This process is deterministic and grounded in formal logic, providing a theoretically consistent guarantee of code correctness.
>
> - **(2) In the code validation phase,** we use a PDDL-based symbolic planner to assess whether a given skill is symbolically feasible under the current observation. This module is based on classic planning theory and provides a formal mechanism to validate the feasibility of executing skills.
>
> Rather than executing LLM-generated code directly, NeSyRo integrates symbolic tools to ensure both logical and environmental consistency, forming a hybrid structure with theoretical grounding. Empirical evaluation remains the standard in LLM-based embodied planning, as seen in works like ZSP [3], SayPlan [4], RT-2 [5], and OpenVLA [6], which validate effectiveness primarily through experiments. In this context, NeSyRo goes further by combining symbolic reasoning with formal verification, providing both theoretical rigor and empirical strength.
>
> ---
> ### W4: Unclear whether baseline selection is comprehensive
> To comprehensively evaluate NeSyRo, we selected baselines that support comparison along three axes central to our framework.
>
> - First, by comparing against CaP, we assess whether NeSyRo improves reliability under partial observability, where LLM-generated policy code lacks full environmental grounding.
>
> - Second, by including CaP w/ Lemur and CodeSift, we test whether existing symbolic Verification & Validation methods, designed for static code analysis, can be directly applied to robot control code. This reveals their limitations in dynamic environments where grounding and safe exploration are essential.
>
> - Third, by comparing against replanning-based methods such as LLM-Planner and AutoGen, which revise the policy only after failure, we examine whether NeSyRo's pre-execution validation leads to safer and more efficient task completion.
>
> This selection ensures that NeSyRo is evaluated against both CaP methods and approaches that incorporate symbolic correctness checks. Similar baselines have been adopted in recent work. For example, planning frameworks [7, 8] have compared against CaP, LLM-Planner, and AutoGen, while neuro-symbolic verification systems [9, 10] have employed Lemur and other formal toolchains to assess the correctness of LLM-generated code. These precedents help situate our choices within the broader research landscape.
>
> ---
> ### Q1: Ablating the recursive element
> To evaluate the impact of the recursive structure in NeSyRo, we conducted an ablation study on long-horizon tasks in the RLBench environment by varying the maximum recursion depth allowed during validation.
>
> |Recursion Depth|0|1|2|3|4|
> |-|-|-|-|-|-|
> |SR|20.0%|30.0%|45.0%|45.0%|45.0%|
> |GC|25.0%|48.2%|52.8%|54.3%|54.0%|
>
> Setting recursion depth to 0 disables interactive validation entirely, equivalent to CaP w/ Lemur, preventing the agent from acquiring missing observations. Depth 1 provides limited probing but cannot resolve multi-step dependencies, resulting in low success rates. With depths of 2 or more, the agent can recursively refine code and acquire missing observations through safe probes, significantly improving performance. Performance improves up to depth 3, then saturates as all missing observations can be recovered within depth 2. These results highlight the importance of recursive verification and validation in NeSyRo.
>
> ---
> ### Q2: Failure modes of the method
> We discussed both quantitative and qualitative failure cases of NeSyRo in Table 3 and Section 4.2 of our paper. Understanding these limitations is critical for guiding future improvements. Beyond the main paper, we summarize some failure cases of NeSyRo as follows:
>
> - **(1) Execution failures due to hardware limitations:**
> In real-world Auxiliary Manipulation tasks (e.g., "place a die into a drawer in a dark room"), NeSyRo sometimes fails to complete the task even when the policy code is correctly generated and symbolically validated. These failures are not caused by errors in perception or planning. Instead, they result from physical limitations, such as the robot's inability to press a light switch accurately. This reflects challenges that originate from hardware rather than the reasoning process.
>
> - **(2) Failures due to limited probe budget:**
> NeSyRo can complete a task by recursively generating safe probe policies until all missing observations are recovered. In principle, unbounded probing would eventually validate all skills and produce a fully grounded policy. However, in real-world deployments, we impose a maximum number of probes to account for time and resource constraints. If this limit is reached before critical observations are obtained, some skills may remain unvalidated, resulting in execution failures. These cases arise from practical constraints rather than limitations of the framework itself.
>
> ---
> We again thank the reviewer. We will incorporate these improvements in the final version.
>
> [1] Younes et al. PPDDL 1.0: An extension to PDDL for expressing planning domains with probabilistic effects. 2004.
>
> [2] De Moura et al. Z3: An Efficient SMT Solver. TACAS 2008.
>
> [3] Huang et al. Language Models as Zero-Shot Planners: Extracting Actionable Knowledge for Embodied Agents. ICML 2022.
>
> [4] Rana et al. SayPlan: Grounding Large Language Models using 3D Scene Graphs for Scalable Robot Task Planning. CoRL 2023.
>
> [5] Zitkovich et al. RT-2: Vision-Language-Action Models Transfer Web Knowledge to Robotic Control. CoRL 2023.
>
> [6] Kim et al. OpenVLA: An Open-Source Vision-Language-Action Model. CoRL 2024.
>
> [7] Yu et al. Language to Rewards for Robotic Skill Synthesis. CoRL 2023.
>
> [8] Choi et al. NeSyC: A Neuro-symbolic Continual Learner For Complex Embodied Tasks In Open Domains. ICLR 2025.
>
> [9] Wu et al. LLM Meets Bounded Model Checking: Neuro-symbolic Loop Invariant Inference. ASE 2024.
>
> [10] Lin et al. VEL: Interactive Formal Verification Environment with Large Language Models via Theorem Proving. NeurIPS 2024.

---

> > ### Comment · Reviewer_rKZ1 · 2025-08-07
> >
> > Thank you for the thorough response. I encourage you to include the recursion ablation in the paper. I believe it supports your method well. I will leave my rating a 5.

---

> > > ### Author Response · Authors · 2025-08-07
> > >
> > > We sincerely appreciate the reviewer for the thoughtful review. We are encouraged by the recognition of our clear contributions and thorough experiments. We agree that the recursion ablation supports our method well, and we include it in the final version.

---

### Note · Authors · 2025-08-11

We thank the reviewers for the constructive discussion. We summarize concerns, responses, and planned revisions.

|Concern|Our Response|
|-|-|
| **Cost and latency** (Reviewer nsGi, UanP) | We reported component-wise metrics and a reliability–cost trade-off via the validation threshold, symbolic calls add overhead to LLM, targeted probes reduce interactions. |
| **Probe safety** (Reviewer nsGi, UanP) | Safe probes are verified and validated like the main policy, LC filters logical risks and CSC contextual risks, irreversible actions remain low. |
| **Assumptions, scope, generalizability** (Reviewer rKZ1, xaVS) | We made explicit the use of symbolic observations under partial observability, deterministic low-level control, and a predefined PDDL domain and skill library; future directions include non-binary LC and MPC-based replanning. |
| **Definitions and clarity** (Reviewer xaVS, nsGi) | One timestep = one skill execution, code skills vs. PDDL actions are distinguished, a formal notion of dynamic environment is given, incl. PDDL contribution to LC. |
| **Baselines** (Reviewer rKZ1, xaVS) | We justified coverage across CaP, verification style, and replanning methods, and clarified Lemur/CodeSift differences and overlap. |
| **Demonstrations** (Reviewer nsGi, xaVS) | We use retrieval of single-skill demonstrations for CSC, ablations show benefit, we term them single-skill demonstrations to avoid confusion with full trajectories. |

Revisions to be included in the final version
- Add recursion ablation supporting our method (Reviewer rKZ1).
- Add cost tables and threshold trade-off analysis, highlighting LLM latency and selective symbolic use (Reviewer nsGi, UanP).
- Expand assumptions/limitations, noting required resources and scope, and non-binary LC with possible MPC for execution failures (Reviewer xaVS).
- Name tools (Z3, Fast Downward) and add verification example with specification snippets (Reviewer xaVS, nsGi).
- Clarify symbolic observations, skill-level timesteps, code vs. PDDL, dynamic environment, LC under partial observations, and PDDL limits (Reviewer xaVS, nsGi).
- Standardize terminology, detail single-skill demonstration retrieval/construction, and report the related ablation (Reviewer nsGi, xaVS).
- Provide baseline rationale and overlap notes with Lemur and CodeSift (Reviewer rKZ1, xaVS).

We hope these clarifications, ablations, and cost analyses help the AC assess our paper, and thank all reviewers and AC for their time and engagement.

---

### Decision · Program_Chairs · 2025-09-17

**Decision:**

Accept (spotlight)

**Comment:**

The paper introduces NeSyRo (Neuro-Symbolic Robotic Task Planning), a framework that extends the Code-as-Policies (CaP) paradigm by adding recursive verification and validation mechanisms to improve the reliability of LLM-generated policies for embodied agents. The framework operates in a loop: an LLM generates task code; a symbolic verification step checks the logical correctness of the code against task specifications; a validation step assesses environmental feasibility via a neuro-symbolic confidence score that combines logical constraints with commonsense priors. If validation fails (confidence is low), the system deploys a safe probe policy to gather additional observations, refining the plan recursively until a high-confidence, executable policy is obtained. Empirical Results show substantial gains in success rate and executability over strong CaP baselines.

**Strengths** The paper addresses a critical bottleneck in LLM-based robotics by improving the reliability of code-generated policies in dynamic environments. The proposed framework combines symbolic verification, neuro-symbolic validation, and safe probing into a recursive loop that ensures robustness. The empirical evaluation spanning RLBench tasks and real-world robotic experiments shows substantial improvements over strong CaP baselines. Ablation studies further validate the necessity of both verification and validation components.

**Weaknesses** Reviewers raised several concerns related to efficiency, assumptions, and scope. The recursive design can slow down planning, which may limit real-time use and make long-horizon tasks challenging. The framework also assumes that probes are always safe and reversible, which might not hold in fragile or irreversible situations. Finally, the reliance on external verification tools raises questions about computational cost.

The authors provide compelling responses, showing that these limitations are reasonable trade-offs for the practical benefits the framework provides. Overall, NeSyRo combines verification, validation, and adaptive probing in a way that is both novel and directly applicable to real-world robotics. Coupled with strong experimental results and a clear, practical system design, the work stands out as highly relevant to the AI and robotics community.